# Modality-Independent Teachers Meet Weakly-Supervised Audio-Visual Event Parser

**Yung-Hsuan Lai,**[1] **Yen-Chun Chen,**[2] **Yu-Chiang Frank Wang**[1,3]

[1]National Taiwan University  [2]Microsoft  [3]NVIDIA

r10942097@ntu.edu.tw, chen.yenchun.tw@gmail.com, frankwang@nvidia.com

## Abstract

Audio-visual learning has been a major pillar of multi-modal machine learning, where the community mostly focused on its *modality-aligned* setting, *i.e.*, the audio and visual modality are *both* assumed to signal the prediction target. With the Look, Listen, and Parse dataset (LLP), we investigate the under-explored *unaligned* setting, where the goal is to recognize audio and visual events in a video with only weak labels observed. Such weak video-level labels only tell what events happen without knowing the modality they are perceived (audio, visual, or both). To enhance learning in this challenging setting, we incorporate large-scale contrastively pre-trained models as the modality teachers. A simple, effective, and generic method, termed **V**isual-**A**udio **L**abel Elab**or**ation (VALOR), is innovated to harvest modality labels for the training events. Empirical studies show that the harvested labels significantly improve an attentional baseline by **8.0** in average F-score (Type@AV). Surprisingly, we found that modality-independent teachers outperform their modality-fused counterparts since they are noise-proof from the other potentially unaligned modality. Moreover, our best model achieves the new state-of-the-art on all metrics of LLP by a substantial margin (**+5.4** F-score for Type@AV). VALOR is further generalized to Audio-Visual Event Localization and achieves the new state-of-the-art as well.[1]

## 1 Introduction

Multi-modal learning has become a pivotal topic in modern machine learning research. Audio-visual learning is undoubtedly one of the primary focuses, as human frequently uses both hearing and vision to perceive the surrounding environment. Countless researchers have devoted to its *modality-aligned* setting with a strong assumption that the audio and visual modality *both* contain learnable clues to the desired prediction target. Numerous audio-visual tasks and algorithms have then been proposed, such as audio-visual speech recognition [1, 66, 68], audio-visual action recognition [22, 53, 81], sound generation from visual data [18, 69, 84], audio-visual question answering [40, 89], and many more. However, almost all real-world events can be audible while invisible, and *vice versa*, depending on how they are perceived. For example, a mother doing dishes in the kitchen might hear a baby crying from the living room, but be unable to directly see what is happening to the baby.

Having observed this potential modality mismatch in generic videos, Tian et al. [73] proposed the Audio-Visual Video Parsing (AVVP) task, which aims to recognize events in videos independently of the audio and visual modalities and also temporally localize these events. AVVP presents an *unaligned* setting of audio-visual learning since all 25 event types considered can be audio-only, visual-only, or audio-visual. Unfortunately, due to the laborious labeling process, Tian et al. [73] created this dataset (Look, Listen, and Parse; LLP) in a weakly-supervised setting.[2] More specifically,

---

[1]Code is available at: `https://github.com/Franklin905/VALOR`.

[2]AVVP, LLP are used interchangeably in the literature. We use AVVP for the task, and LLP for the dataset.

37th Conference on Neural Information Processing Systems (NeurIPS 2023).

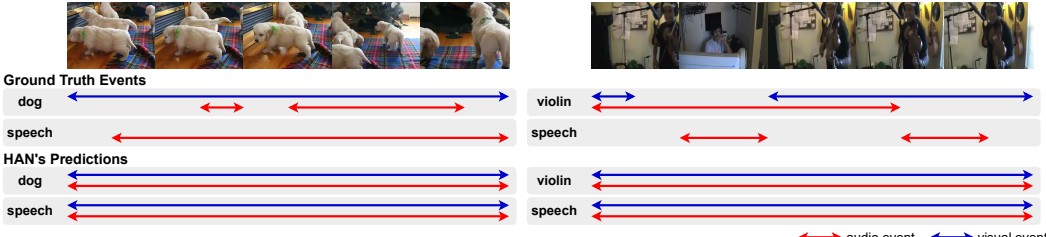

Figure 1: **Modality-unaligned samples from LLP.** Note that recent AVVP approaches like HAN [73] are vulnerable to unaligned data modality and produce incorrect predictions.

only video-level event annotations are available at training. In other words, the modality (audio, visual, or both) and the timestamp of which an event occurs are not given to the learning model.

The AVVP task poses significant challenges from three different perspectives. First, an event is typically modality independent, *i.e.*, knowing an event occurs in one modality says nothing about the other modality. As illustrated in Fig. 1, a sleeping dog is seen but may not be heard; conversely, a violin being played could sometimes go out of the camera view. Second, existing works heavily rely on the Multi-modal Multiple Instance Learning (MMIL) loss [73] to soft-select the modality (and timestamp), given only weak modality-less labels. This would be challenging for models to learn the correct event modality without observing a large amount of data. The uni-modal guided loss via label smoothing is also used to introduce uncertainty to the weak labels and thus regularize modality recognition. However, we hypothesize this improvement could be sub-optimal because no explicit modality information is introduced. Finally, AVVP requires models to predict events for all 1-second segments in a given video. Learning from weak video-level labels without timestamps makes it challenging for models to predict on a per-segment basis.

To address the above challenges in AVVP, we propose to incorporate large-scale pre-trained open-vocabulary models, namely CLIP [56] and CLAP [79], to enhance learning with weak labels. Pre-trained on pixels and waveforms (and contrastively pre-trained with natural language), these models are inherently isolated from potential spurious noise from the other modality. Another benefit is the applicability of their prediction in an open-vocabulary fashion. Therefore, to benefit from CLIP and CLAP, we aim to harvest explicit modality learning signals from them. Moreover, we aim to inference these models per video segment, yielding fine-grained temporal annotations.

While it might be tempting to naively treat these pre-trained models as teachers and then applies knowledge distillation (KD) [28], this could be sub-optimal as some events are difficult to distinguish from a single modality, even for humans. For example, cars, motorcycles, and lawn mowers all produce similar sounds. To better utilize CLIP and CLAP, we introduce **V**isual-**A**udio **L**abel Elab**or**ation (VALOR), to harvest modality and timestamp labels in LLP. We prompt CLIP/CLAP with natural language description of all visual/audio event types for each video segment-by-segment and then extract labels when a threshold is met. Additionally, implausible events are filtered out using the original weak labels accompanied with the video to mitigate the above indistinguishable problem. VALOR constructs fine-grained temporal labels in both modalities so that models have access to explicit training signals.

In addition to achieving the promising performance of AVVP, we observe that modality-independent teachers, CLIP and CLAP, generate more reliable labels than a modality-fused one, a cross-modal transformer. We also showcase the generalization capability of VALOR via the Audio-Visual Event Localization (AVE) task, in which our method also achieves the new state-of-the-art. Our contributions are summarized as follows:

- A simple and effective AVVP framework, VALOR, is proposed to harvest modality and temporal labels directly from video-level annotations, with an absolute improvement of **+8.0** F-score.

- We are the first to point out that modality independence could be crucial for audio-visual learning in the *unaligned* and weakly-supervised setup.

- Our VALOR achieves new state-of-the-art results with significant improvements on AVVP (**+5.4** F-score), with generalization to AVE (**+4.4** accuracy) jointly verified.

## 2 Preliminaries

**Audio-Visual Video Parsing (AVVP)** The AVVP [73] task is to recognize events of interest in a video in both visual and audio modalities and to temporally identify the associated frames. For the benchmark dataset of Look, Listen, and Parse (LLP), a $T$-second video is split into $T$ non-overlapping segments. Each video segment is paired with a set of multi-class event labels $(\boldsymbol{y}_t^v, \boldsymbol{y}_t^a) \in \{0, 1\}^C$ ($\boldsymbol{y}_t^v$: visual events, $\boldsymbol{y}_t^a$: audio events, $C$: number of event types). However, in the training split, the dense 'segment-level' labels $(\boldsymbol{y}_t^v, \boldsymbol{y}_t^a)$ are *not* available. Instead, only the global modality-less 'video-level' labels $\boldsymbol{y} := \max_t \{\boldsymbol{y}_t^v \wedge \boldsymbol{y}_t^a\}_{t=1}^T$ are provided ($\wedge$: element-wise 'logical and'). In other words, AVVP models need to be learned in a weakly-supervised setting.

**Baseline Model** We now briefly review the model of Hybrid Attention Network (HAN) [73], which is a common baseline for AVVP. In HAN, ResNet-152 [26] and R(2+1)D [75] are employed to extract 2D and 3D visual features. Subsequently, they are concatenated and projected into segment-level features $\boldsymbol{F}^v = \{\boldsymbol{f}_t^v\}_{t=1}^T \in \mathbb{R}^{T \times d}$ ($d$: hidden dimension). Segment-level audio features $\boldsymbol{F}^a = \{\boldsymbol{f}_t^a\}_{t=1}^T \in \mathbb{R}^{T \times d}$ are extracted using VGGish [27] and projected to the same dimension. HAN takes these features and aggregates the intra-modal and cross-modal information through self-attention and cross-attention:

$$\tilde{\boldsymbol{f}}_t^a = \boldsymbol{f}_t^a + \text{Att}(\boldsymbol{f}_t^a, \boldsymbol{F}^a, \boldsymbol{F}^a) + \text{Att}(\boldsymbol{f}_t^a, \boldsymbol{F}^v, \boldsymbol{F}^v) \tag{1}$$

$$\tilde{\boldsymbol{f}}_t^v = \boldsymbol{f}_t^v + \text{Att}(\boldsymbol{f}_t^v, \boldsymbol{F}^v, \boldsymbol{F}^v) + \text{Att}(\boldsymbol{f}_t^v, \boldsymbol{F}^a, \boldsymbol{F}^a), \tag{2}$$

where $\text{Att}(\boldsymbol{q}, \boldsymbol{K}, \boldsymbol{V})$ denotes multi-head attention [76]. Following Transformer's practice, the outputs are further fed through LayerNorms [6] and a 2-layer FFN to yield $\hat{\boldsymbol{f}}_t^a, \hat{\boldsymbol{f}}_t^v$. With another linear layer, the hidden features are transformed into categorical logits $\boldsymbol{z}_t^v, \boldsymbol{z}_t^a$ for visual and audio events, respectively. Finally, the segment-level audio and visual event categorical probabilities, $\boldsymbol{p}_t^a$ and $\boldsymbol{p}_t^v$ ($\in [0, 1]^C$), are obtained by applying Sigmoid activation.

As a key module in Tian et al. [73], Multi-modal Multiple Instance Learning pooling (MMIL) is applied to address the above weakly-supervised learning task, which predicts the audio and visual event probabilities ($\boldsymbol{p}^m$, $m \in \{a, v\}$, audio and visual modalities) as:

$$\boldsymbol{A}^m = \{\boldsymbol{\alpha}_t^m\}_t = \text{softmax}_t(\hat{\boldsymbol{F}}^m \boldsymbol{W}^m), \qquad \boldsymbol{p}^m = \sum_t \boldsymbol{\alpha}_t^m \odot \boldsymbol{p}_t^m, \tag{3}$$

where trainable parameters $\boldsymbol{W}^m \in \mathbb{R}^{d \times C}$ are implemented as linear layers ($\odot$: element-wise product). For video-level event probability $\boldsymbol{p}$:

$$\mathbf{B} = \{\{\boldsymbol{\beta}_t^m\}_t\}_m = \text{softmax}_m(\hat{\mathbf{X}} \boldsymbol{W}), \qquad \boldsymbol{p} = \sum_m \sum_t \boldsymbol{\beta}_t^m \odot \boldsymbol{\alpha}_t^m \odot \boldsymbol{p}_t^m, \tag{4}$$

where $\hat{\mathbf{X}} = \{\hat{\boldsymbol{F}}^m\}_m \in \mathbb{R}^{2 \times T \times d}$ and $\boldsymbol{W}$ as a trainable linear layer. Moreover, modality training targets are obtained via label smoothing (LS) [71]: $\tilde{\boldsymbol{y}}^m = \text{LS}(\boldsymbol{y})$. Finally, the model is trained with binary cross entropy (BCE) as the loss function:

$$\mathcal{L}_{\text{base}} = \mathcal{L}_{\text{video}} + \mathcal{L}_{\text{guided}}^a + \mathcal{L}_{\text{guided}}^v, \quad \mathcal{L}_{\text{video}} = \text{BCE}(\boldsymbol{p}, \boldsymbol{y}), \quad \mathcal{L}_{\text{guided}}^m = \text{BCE}(\boldsymbol{p}^m, \tilde{\boldsymbol{y}}^m). \tag{5}$$

In summary, by the attention mechanisms introduced in HAN, MMIL pooling assigns event labels for each modality across time segments with only video-level event labels observed during training.

## 3 Method

With only video-level event labels observed during training, we address three major challenges of AVVP: 1) modality independence of events' occurrence, 2) reliance on MMIL pooling for event label assignment under insufficient data, and 3) demand for dense temporal predictions. To address these challenges, we propose to leverage large-scale pre-trained contrastive models, CLIP and CLAP, to extract modality-aware, temporally dense training signals to guide model learning.

### 3.1 Zero-Shot Transfer of Contrastive Pre-trained Models

Radford et al. [56] proposed Contrastive Language-Image Pre-training (CLIP) to utilize web-scale image-text pairs to train a strong image encoder. As a result, CLIP overthrows the limitation of predicting predefined categories. Due to its large training data size (400M), CLIP has demonstrated remarkable zero-shot performance on a wide range of visual recognition tasks. All the above motivates us to incorporate CLIP to improve visual event recognition in AVVP.

In our work, CLIP's visual understanding of AVVP is extracted as the following. We extract $T$ evenly spaced video frames and pass them into CLIP's image encoder to obtain the visual features $\{\boldsymbol{f}_t^{\text{CLIP}}\}_{t=1}^T \in \mathbb{R}^{T \times d_2}$ ($d_2$: the dimension of CLIP's feature). For simplicity and readability, we will

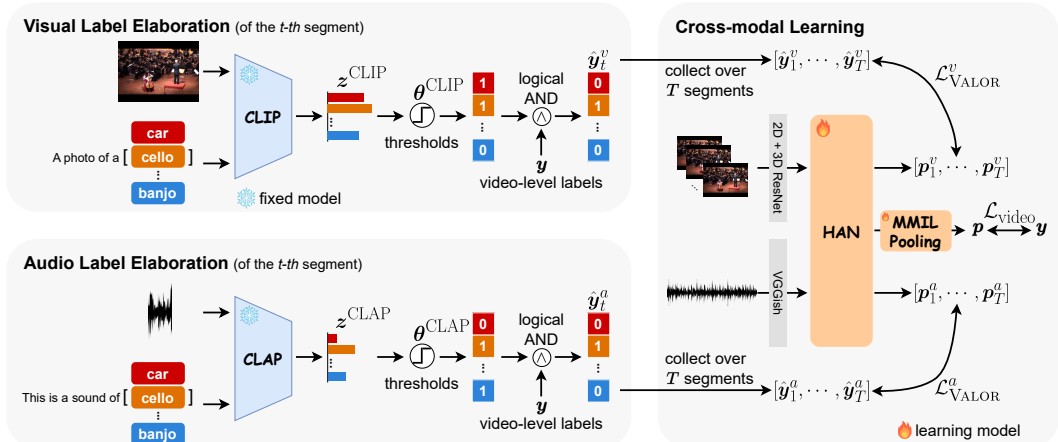

Figure 2: **VALOR framework.** With modality-independent label elaboration via CLIP and CLAP, the harvested temporally dense labels serve as additional modality- and time-aware cues.

omit the time subscript $t$ for the remainder of this paper when there is no ambiguity. Next, we convert the AVVP event categories to concepts that CLIP understands. A caption for each event is constructed by adding a "A photo of" prefix to the event's natural language form. These captions are processed by the CLIP's text encoder, resulting in event features $G^{\text{CLIP}} = \{g_c^{\text{CLIP}}\}_{c=1}^C \in \mathbb{R}^{C \times d_2}$, where $c$ indexes the events, and $g_c$ represents the text feature of the $c$-th event. Frame-level event logits $z^{\text{CLIP}} \in \mathbb{R}^C$ can be obtained by calculating the inner products:

$$z^{\text{CLIP}} = f^{\text{CLIP}} G^{\text{CLIP}\top}. \tag{6}$$

In light of the notable success of CLIP [56], several studies have sprouted to research on learning representative audio embeddings and text embeddings through Contrastive Language-Audio Pre-training (CLAP) [13, 15, 25, 49, 79]. In the same way as images and text are encoded in CLIP, web-scale audios and text are pre-trained with a contrastive objective in CLAP. Symmetrically, we obtain CLAP's understanding of AVVP audios as the following. From the audio channel, the raw waveform is extracted and split into $T$ segments with the same lengths and then fed into CLAP, yielding segment-level audio features $\{f_t^{\text{CLAP}}\}_t \in \mathbb{R}^{T \times d_3}$ ($d_3$: the dimension of CLAP's feature). On the other hand, an audio event caption is constructed by adding the prefix "This is a sound of" to each AVVP event's name. Processed by the CLAP text encoder, we obtain $G^{\text{CLAP}} = \{g_c^{\text{CLAP}}\}_c \in \mathbb{R}^{C \times d_3}$. Segment-level audio event logits $z^{\text{CLAP}} \in \mathbb{R}^C$ are obtained by the inner products:

$$z^{\text{CLAP}} = f^{\text{CLAP}} G^{\text{CLAP}\top}. \tag{7}$$

We note that Eqn. (6) and (7) can be viewed as CLIP's and CLAP's understanding of the associate video frame in the event space of AVVP.

### 3.2 Harvesting Training Signals

Given the logits $z^{\text{CLIP}}$ and $z^{\text{CLAP}}$, we aim to convert them to useful training signals for the AVVP task. An intuitive idea is to teach our model via knowledge distillation (KD) [28]. To deploy KD in training, segment-level normalized probabilities are first computed: $q^P = \text{softmax}_c(z^P)$, $q^m = \text{softmax}_c(z^m)$, where $(m, P) \in \{(v, \text{CLIP}), (a, \text{CLAP})\}$ denotes data **m**odality (audio/visual) and **p**re-trained model (CLIP/CLAP) pair. Next, KL-divergence for all segments is calculated: $\mathcal{L}_{\text{KD}}^m = \sum_t \text{KL}(q_t^P, q_t^m)$. Finally, KD training is done by optimizing the loss function:

$$\mathcal{L}_{\text{KD}} = \mathcal{L}_{\text{video}} + \mathcal{L}_{\text{KD}}^a + \mathcal{L}_{\text{KD}}^v. \tag{8}$$

However, as we find out empirically (shown in Table 4), this is *not* the optimal usage of CLIP and CLAP. We hypothesize that some events are hard to distinguish from a single modality, *e.g.* cars, motorcycles, and lawn mowers produce the sound of an engine. Therefore, we design VALOR, utilizing video-level labels to filter out the impossible events, hence mitigating the confusion.

**Visual-Audio Label Elaboration (VALOR)** To better exploit CLIP and CLAP, we design a simple yet effective method, VALOR, to harvest dense labels in both modalities. In particular, we first define class-dependent thresholds $\theta^P \in \mathbb{R}^C$ for each modality to obtain segment-level labels from logits. Next, the impossible events are excluded using the given video-level labels, done via logical AND. Formally, this process can be written as: $\hat{y}_t^m = \{z_t^P > \theta^P\} \wedge y$, with the overall loss function:

$$\mathcal{L}_{\text{VALOR}} = \mathcal{L}_{\text{video}} + \mathcal{L}_{\text{VALOR}}^a + \mathcal{L}_{\text{VALOR}}^v, \quad \mathcal{L}_{\text{VALOR}}^m = \sum_t \text{BCE}(p_t^m, \hat{y}_t^m). \tag{9}$$

To summarize, we design a simple yet effective method, VALOR, to utilize large-scale pre-trained contrastive models, CLIP and CLAP, to generate segment-level labels in both modalities. Due to the nature of immunity to spurious noise from the other modality, the contrastive pre-training methods, and the large pre-training dataset size, CLIP and CLAP are able to provide reliable labels in visual and audio modality, respectively. In addition, they are able to provide temporally dense labels to explicitly guide the model in learning events in each segment.

# 4 Related Work

## 4.1 Audio-Visual Video Parsing with Look, Listen, and Parse

For AVVP, research flourishes along two orthogonal directions: enhancing the model architecture and label refinement. Architectural improvements include cross-modal co-occurrence module [45], class-aware uni-modal features and cross-modal grouping [51], and Multi-Modal Pyramid attention [87]. On the other hand, label refinement shares a similar spirit with ours. MA [77] corrupted the data by swapping the audio channel of two videos with disjoint video-level event sets. The model's likelihood of the corrupted data was then used to determine the modality label. More recently, JoMoLD [11] utilized a two-stage approach. First, an AVVP model was trained as usual. Next, another model was trained while denoising the weak labels with prior belief from the first model. Both MA and JoMoLD produced global modality labels without timestamps. Concurrent to ours, VPLAN [96] and LSLD [16] generate dense temporal visual annotations with CLIP; however, the audio labels remain absent. Our VALOR represents a *unified* framework to elaborate the weak labels, along modality *and* temporal dimension, via zero-shot transfer of pre-trained models. We further emphasize the importance of modality independence when synthesizing modality supervision.

## 4.2 More Audio-Visual Learning

**Audio-Visual Event Localization (AVE)** Tian et al. [72] proposed AVE to recognize the audio-visual event in a video while localizing its temporal boundaries. Numerous studies have been conducted, including Lin et al. [46] with seq2seq models, Lin and Wang [44] using intra&inter frame Transformers, Wu et al. [78] via dual attention matching, audio-spatial channel-attention by Xu et al. [82], positive sample propagation from Zhou et al. [95], and Xia and Zhao [80] employing background suppression. We generalize VALOR to AVE's weakly supervised setting.

**Audio-Visual Assistance** While significant advancements have been made in speech recognition, speech enhancement, and action recognition, noise or bias residing in the uni-modal data is still problematic. An effective solution could involve integrating data from an additional modality. This research direction encompasses various areas including speech recognition [1, 30, 66, 68], speaker recognition [12, 14, 55, 61, 63, 67], action recognition [22, 35, 36, 53, 81], speech enhancement or separation [2, 3, 34, 39, 50, 60], and object sound separation [7, 20, 21, 59, 74, 83, 91, 92].

**Audio-Visual Correspondence and Understanding** Humans possess an impressive capacity to deduce occurrences in one sensory modality using information solely from another. This fascinating human ability to perceive across modalities has inspired researchers to delve into sound generation from visual data [18, 19, 37, 45, 54, 69, 84, 93], video generation from audio [38, 41, 43, 90, 94], and audio-visual retrieval [42, 70]. In the pursuit of understanding how humans process audio-visual events, numerous studies have been undertaken on audio-visual understanding tasks such as sound localization in videos [5, 31, 32, 52, 64], audio-visual navigation [8–10, 17, 48, 86, 88], and audio-visual question answering [4, 24, 29, 40, 62, 65, 89].

# 5 Experiments

## 5.1 Experimental Setup

**Dataset and Metrics** The LLP dataset is composed of 11849 10-second Youtube video clips covering 25 event categories, such as human activities, musical instruments, vehicles, and animals. The dataset is divided into training, validation, and testing splits, containing $10,000$, $649$, and $1200$ clips, respectively. The official evaluation uses F-score to evaluate audio (A), visual (V), and

Table 1: **AVVP benchmark.** Note that pseudo label denoising is not applied for VPLAN[†]. VALOR+ is trained on a thinner yet deeper HAN of similar size. VALOR++ further uses CLIP and CLAP as feature extractors and significantly boosts all metrics. The best numbers are in bold and the second best numbers are underlined.

| Methods | Segment-level | | | | | Event-level | | | | |
|---|---|---|---|---|---|---|---|---|---|---|
| | A | V | AV | Type | Event | A | V | AV | Type | Event |
| AVE [72] | 47.2 | 37.1 | 35.4 | 39.9 | 41.6 | 40.4 | 34.7 | 31.6 | 35.5 | 36.5 |
| AVSDN [46] | 47.8 | 52.0 | 37.1 | 45.7 | 50.8 | 34.1 | 46.3 | 26.5 | 35.6 | 37.7 |
| HAN [73] | 60.1 | 52.9 | 48.9 | 54.0 | 55.4 | 51.3 | 48.9 | 43.0 | 47.7 | 48.0 |
| MM-Pyr [87] | 60.9 | 54.4 | 50.0 | 55.1 | 57.6 | 52.7 | 51.8 | 44.4 | 49.9 | 50.5 |
| MGN [51] | 60.8 | 55.4 | 50.4 | 55.5 | 57.2 | 51.1 | 52.4 | 44.4 | 49.3 | 49.1 |
| CVCMS [47] | 59.2 | 59.9 | 53.4 | 57.5 | 58.1 | 51.3 | 55.5 | 46.2 | 51.0 | 49.7 |
| DHHN [33] | 61.3 | 58.3 | 52.9 | 57.5 | 58.1 | 54.0 | 55.1 | 47.3 | 51.5 | 51.5 |
| MA [77] | 60.3 | 60.0 | 55.1 | 58.9 | 57.9 | 53.6 | 56.4 | 49.0 | 53.0 | 50.6 |
| JoMoLD [11] | 61.3 | 63.8 | 57.2 | 60.8 | 59.9 | 53.9 | 59.9 | 49.6 | 54.5 | 52.5 |
| VPLAN[†] [96] | 60.5 | 64.8 | 58.3 | 61.2 | 59.4 | 51.4 | 61.5 | 51.2 | 54.7 | 50.8 |
| VALOR | 61.8 | 65.9 | 58.4 | 62.0 | 61.5 | 55.4 | 62.6 | 52.2 | 56.7 | 54.2 |
| VALOR+ | 62.8 | 66.7 | 60.0 | 63.2 | 62.3 | 57.1 | 63.9 | 54.4 | 58.5 | 55.9 |
| VALOR++ | **68.1** | **68.4** | **61.9** | **66.2** | **66.8** | **61.2** | **64.7** | **55.5** | **60.4** | **59.0** |

audio-visual (AV) events separately. Type@AV (Type) is the averaged F-scores of A, V, and AV. Event@AV (Event) measures the ability to detect events in both modalities by combining audio and visual event detection results. Different from segment-level metrics, the event-level metrics treat consecutive positive segments as a whole, and mIoU of $0.5$ is applied to calculate F-scores.

**Implementation Details**  Unless otherwise specified, VALOR uses HAN under a fair setting w.r.t. previous works with same data pre-processing. For the visual feature extraction, video frames are sampled at $8$ frames per second. Additionally, we conduct experiments using CLIP and CLAP as feature extractors. The pre-trained ViT-L CLIP and HTSAT-RoBERTa-fusion CLAP are used to generate labels and extract features. Note that for all experiments with CLAP, we use the implementation from Wu et al. [79] pre-trained on LAION-Audio-630K. We do not use the version pre-trained on AudioSet (a larger pre-training corpus) since it overlaps with the AVVP validation and testing videos.

## 5.2  Unified Label Elaboration for State-of-the-Art Audio-Visual Video Parsing

To demonstrate the effectiveness of VALOR, we evaluate our method on the AVVP benchmark. Existing works include: 1) weakly-supervised audio-visual event localization methods AVE and AVSDN, 2) HAN and its network architecture advancements MM-Pyramid, MGN, CVCMS, and DHHN, and 3) different label refinement methods MA, JoMoLD, and VPLAN. We report the results on the test split of the LLP dataset in Table 1.

We achieve the new state-of-the-art (SOTA) on all metrics consistently with a large margin. Our method VALOR significantly improves the baseline (HAN) by **8.0** in segment-level Type@AV. Compared to previous published SOTA, JoMoLD, VALOR scores higher on all metrics, including the **5.4** F-score improvement for segment-level Type@AV, under a fair setting. With light hyper-parameter tuning, VALOR+ further achieves a significant $2.4$ improvement on Type@AV, with a deeper yet thinner HAN while keeping a similar number of trainable parameters. Our improvement on the audio side w.r.t. the concurrent preprint VPLAN is more significant than the visual side, which may be attributed to our effective audio teacher CLAP and label elaboration along the modality axis. We empirically conclude that VALOR has successfully unified label refinement along both modality and temporal dimensions. To push to the limits, we further proposed VALOR++ by replacing the feature extraction models with CLIP and CLAP, achieving another consistent boost, including 3.0 in segment Type@AV. We will release the VALOR++ pre-trained checkpoint, features, and harvested labels to boost future AVVP research.

Table 2: **Selection of modality-independent labeler.** Note that utilizing a cross-modal labeler HAN instead of CLIP and CLAP to generate segment-level labels hardly improves the baseline (HAN). On the other hand, modality-less segment-level labels deteriorates the performance. All results are reported on the **validation** split of LLP.

| Dense Labeler | Modality Label | Segment-level | | | | | Event-level | | | | |
|---|---|---|---|---|---|---|---|---|---|---|---|
| | | A | V | AV | Type | Event | A | V | AV | Type | Event |
| None | ✔ | 62.0 | 54.5 | 50.2 | 55.6 | 57.1 | 53.5 | 50.5 | 43.6 | 49.2 | 50.3 |
| HAN | ✔ | 62.1 | 56.4 | 52.1 | 56.8 | 57.6 | 53.4 | 52.0 | 45.4 | 50.3 | 50.6 |
| CLIP&CLAP | ✘ | 41.0 | 59.0 | 34.5 | 44.9 | 52.1 | 33.2 | 56.2 | 28.2 | 39.2 | 43.1 |
| CLIP&CLAP | ✔ | **62.7** | **66.3** | **61.0** | **63.4** | **61.8** | **55.5** | **62.0** | **54.1** | **57.2** | **53.8** |

Table 3: **Fidelity of the elaborated labels.** We conduct a comparison between the segment-level labels generated from VALOR and those from a naive approach where we assume video-level labels also serve as segment-level labels. We directly evaluate these pseudo labels on the validation split before training. The results clearly indicate that VALOR-generated labels are more accurate than the naive ones.

| Label Generation Methods | Audio | Visual | Audio-Visual |
|---|---|---|---|
| Video Labels | 80.08 | 67.21 | 59.45 |
| VALOR | **85.07** (+4.99) | **82.14** (+14.93) | **77.07** (+17.62) |

## 5.3 Ablation Studies

The impressive results achieved in Table 1 are based on careful design. In this subsection, we elaborate on why we choose CLIP and CLAP to synthesize dense labels with modality annotations with empirical support. Furthermore, we break down the loss function and modeling components into orthogonal pieces and evaluate their individual effectiveness.

**How to choose the labeler?** In Table 2, we show the necessity of modality-independent pre-trained models (CLIP and CLAP) over the multi-modal model (HAN) as the labeler (2nd row) and that modality-aware labels beat modality-agnostic labels (3rd row). We aim to demonstrate the necessity and importance of **using large-scale pre-trained uni-modal models** to annotate **modality-aware segment-level labels**. To validate the former, we employ a baseline model (HAN) that has been trained on AVVP to individually annotate segment-level labels within the two modalities. Experimental results show that modality-aware temporal dense labels generated by a multi-modal model (HAN), learned from weak labels, are less effective than those generated by large-scale pre-trained uni-modal models (CLIP and CLAP), thereby underscoring the essentiality of using large-scale pre-trained uni-modal models. Subsequently, to validate the latter, we generate modality-agnostic segment-level labels from CLIP and CLAP, meaning that these labels only reveal the events occurring in each segment but do not disclose the modality of the event. As seen from the third row of Table 2, while such a labeling method increases the F-score for visual events, it dramatically decreases the F-score for audio events. The overall performance (Type F-score) is even worse than that of HAN (the first row), clearly indicating the importance of modality-aware labeling for the model to learn the AVVP task effectively.

**How accurate are the elaborated labels?** To measure the fidelity of the pseudo labels generated via VALOR in audio and visual modalities, we conduct a comparison between the segment-level labels generated from VALOR and those from a naive approach where we assume that video-level labels also serve as segment-level labels, *i.e.*, we assume that an event occurs in both modalities and all segments if it occurs in the video. We directly evaluate these pseudo labels on the validation split before using them for training. The results, presented in Table 3, clearly demonstrate the superiority of our generated segment-level audio and visual pseudo labels compared to the naive counterparts. Notably, our segment-level visual F-score exceeds the naive approach by nearly 15 points while the audio-visual F-score significantly surpasses for more than 17 points. These results highlight the reliability of the VALOR-generated pseudo labels, which provide more faithful temporal and modal information to facilitate model training.

Table 4: **Ablation study.** "global" denotes only video-level labels observed, while "dense" indicates segment-level labels available as ground truth. "base" is the baseline method [73]. "New Feat." denotes the use of features from CLAP, CLIP, and R(2+1)D, and "Deep HAN" is that of the 256-dim 4-layer HAN model. All results are reported on the **validation** split of LLP.

| Audio Loss | | Visual Loss | | New | Deep | Segment-level | | | | |
| global | dense | global | dense | Feat. | HAN | A | V | AV | Type | Event |
|---|---|---|---|---|---|---|---|---|---|---|
| base | ✗ | base | ✗ | ✗ | ✗ | 62.0 | 54.5 | 50.2 | 55.6 | 57.1 |
| ✗ | KD | ✗ | KD | ✗ | ✗ | 51.1 | 64.0 | 48.0 | 54.3 | 55.5 |
| VALOR | ✗ | VALOR | ✗ | ✗ | ✗ | 62.1 | 65.8 | 59.0 | 62.3 | 61.2 |
| base | ✗ | ✗ | VALOR | ✗ | ✗ | 60.5 | 66.7 | 60.8 | 62.7 | 59.8 |
| ✗ | VALOR | base | ✗ | ✗ | ✗ | 62.2 | 54.5 | 52.7 | 56.5 | 56.5 |
| ✗ | VALOR | ✗ | VALOR | ✗ | ✗ | 62.7 | 66.3 | 61.0 | 63.4 | 61.8 |
| ✗ | VALOR | ✗ | VALOR | ✗ | ✔ | 64.5 | 67.1 | 63.1 | 64.9 | 63.2 |
| ✗ | VALOR | ✗ | VALOR | ✔ | ✔ | **71.4** | **69.4** | **64.9** | **68.6** | **69.7** |

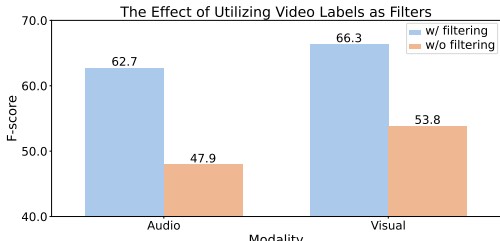

Figure 3: **Ablation study of whether using video-level labels as filters.**

Figure 4: **The extent to which the models address the modality non-alignment issue.**

**How to use the elaborated labels?** We conduct an ablation study on utilizing CLIP and CLAP together. The results are presented in Table 4. The replacement of the smoothed video-level event labels $\tilde{y}^a$ and $\tilde{y}^v$ with their respective refined weak labels $\hat{y}^a$ and $\hat{y}^v$ derived from our method leads to a significant increase in the Type@AV F-score, from $54.0$ to $60.8$. This finding underscores the importance of incorporating labels that are proximal to ground truth, albeit weak. Furthermore, we leverage the CLIP and CLAP models to generate segment-level labels for each modality. This approach results in an improvement of $8.0$ Type@AV F-score over the baseline, indicating that explicitly informing the model of the events occurring in each segment of the audio-visual video facilitates the learning of the Audio-Visual Video Parsing (AVVP) task. In addition, CLIP and CLAP are also used to obtain more representative features. Replacing the ResNet-152 and VGGish features with CLIP and CLAP features yields a Type@AV F-score improvement of $4.0$.

**Whether using video-level labels as filters?** Video-level labels are pivotal for generating reliable pseudo labels in our method, where we employ them as filters to eliminate impossible events misclassified by CLIP or CLAP. In Figure 3, we conduct experiments to underscore the necessity of using video-level labels as filters. Notably, without utilizing video-level labels as filters, both audio and visual F-scores plummet, reaching $47.9$ and $53.8$, respectively.

**How well can the modality non-aligned problem be solved?** As we have pointed out that the modality independence of events is one of the crucial challenges in the AVVP task, we assess the extent of the modality non-aligned problem in the LLP dataset and the extent to which the models can solve the problem. First, we define the word "segment-level event" as the cumulative sum of the number of events that occur without modality differences across all segments. In other words, if an audio event and a visual event from the same category occur within a segment, they are counted as a single "segment-level event." In the LLP dataset's validation split, there are $9126$ segment-level events. Among these, $4048$ segment-level events are modality non-aligned, *i.e.*, they occur in exactly one modality. To measure how well trained models address the modality non-aligned issue, we conduct experiments involving several models, including our own, to predict both the modality and event of these segments. A successful prediction entails correctly identifying the event and confirming its presence in both modalities. The results, as displayed in Figure 4, reveal that HAN exhibits the poorest performance in predicting modality non-aligned events. Conversely, our methods VALOR and VALOR++ outperform the prior SOTA, JoMoLD. This highlights the effectiveness of our approach in mitigating the modality non-alignment challenge within the AVVP task.

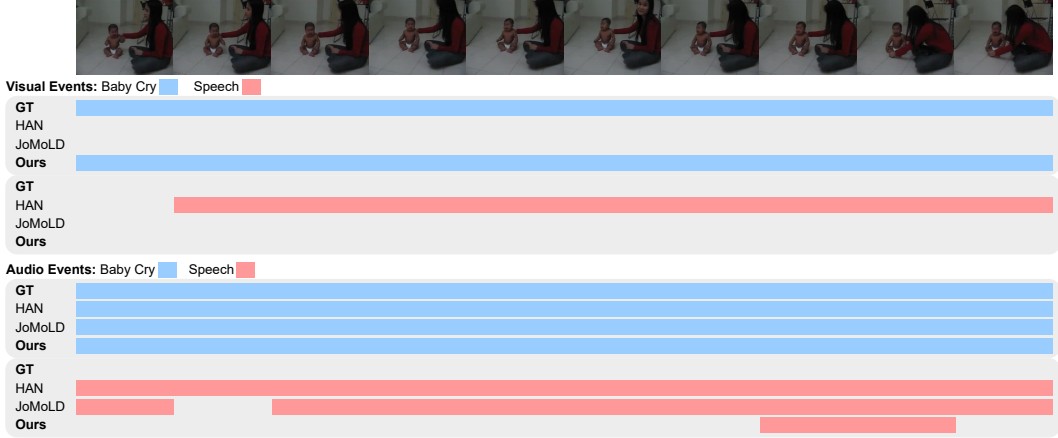

Figure 5: **Qualitative Comparison with Previous AVVP Works.** "GT" denotes the ground truth annotations. We compare with HAN [73] and JoMoLD [11].

## 5.4 Generalize VALOR to Audio-Visual Event Localization

In this section, we showcase the additional generalization ability of VALOR by applying it to the Audio-Visual Event Localization (AVE) task. We consider the weakly-supervised version of AVE, where segment-level ground truths are not available to the model during training, meaning that no timestamp is provided for the event, motivating us to apply VALOR to harvest event labels. Without task-specific modification, we directly apply HAN and VALOR to AVE. The only difference is that at inference, we combine the audio and visual prediction to obtain the audio-visual event required in this task. Please refer to the supplementary for more implementation details of the AVE task.

**Quantitative Results** From Table 5, we observe that our baseline method performs on par with the previous state-of-the-art method CMBS [80]. When our method is applied to the model, the accuracy leaps from 75.3 to 80.4, indicating the generalizability of our method. In addition, we surpass CMBS [80] and have become the new state-of-the-art on the weakly-supervised AVE task with an improvement of **4.4** in accuracy.

Table 5: Results on the AVE task.

| Method | Accuracy(%) |
| --- | --- |
| *VGG-like, VGG-19 features* | |
| AVEL [72] | 66.7 |
| AVSDN [46] | 67.3 |
| CMAN [85] | 70.4 |
| AVRB [58] | 68.9 |
| AVIN [57] | 69.4 |
| AVT [44] | 70.2 |
| CMRAN [82] | 72.9 |
| PSP [95] | 73.5 |
| CMBS [80] | 74.2 |
| *VGG-like, Res-151 features* | |
| AVEL [72] | 71.6 |
| AVSDN [46] | 74.2 |
| CMRAN [82] | 75.3 |
| CMBS [80] | 76.0 |
| *CLAP, CLIP, R(2+1)D features* | |
| HAN | 75.3 |
| VALOR | **80.4** |

## 5.5 Additional Analyses

**Qualitative Comparison** Aside from quantitative comparison with previous AVVP works, we perform a qualitative evaluation as well. We qualitatively compare with the baseline method HAN [73] and the state-of-the-art method JoMoLD [11]. From Figure 5, it can be seen that only our model can correctly predict the "Baby Cry" visual event. HAN not only fails to predict "Baby Cry" correctly but also mistakenly identifies the woman in the video as speaking. In the audio modality, all models correctly predict the presence of "Baby Cry" in the sound, but they also simultaneously misinterprets that someone is talking. Among all models, our model makes the least severe misjudgments."

**Class-wise F-score Comparison.** We further evaluate the effectiveness of providing accurate uni-modal segment-level pseudo labels for the model training. We visualize class-wise improvements between our generated segment-level labels for each modality and the naive segment-level labels derived from the video-level labels. In Figure 6, we observe that when our audio pseudo labels are used, most of the audio events improve. In Figure 7, when our visual pseudo labels are used, nearly every event's F-score increases. These results indicate the effectiveness of our method in guiding the model to learn events in each modality. For the inferior performance on the "Speech" event, since CLIP is inept at extracting fine-grained visual information, it is not expected to recognize the "Speech" event well, which requires close attention on mouth movements.

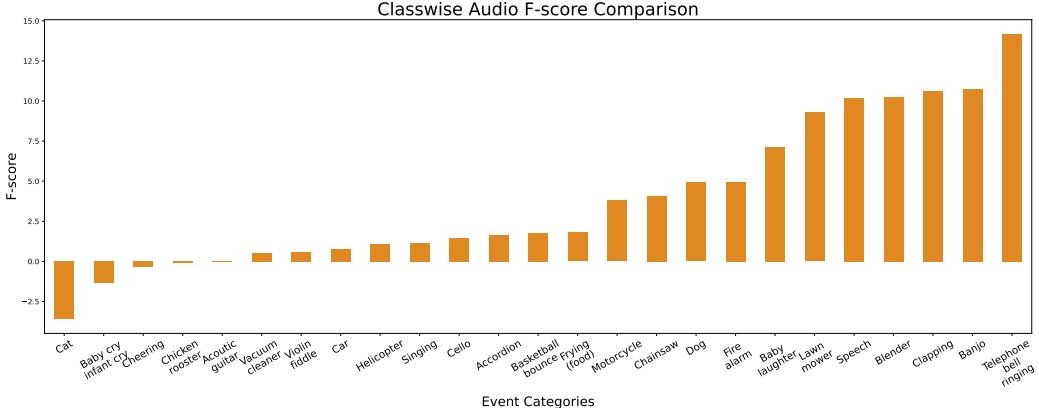

Figure 6: **Class-wise improvement on audio events.** Using the derived audio segment-level pseudo label is advantageous over the baseline using video-level labels as if they were audio segment-level.

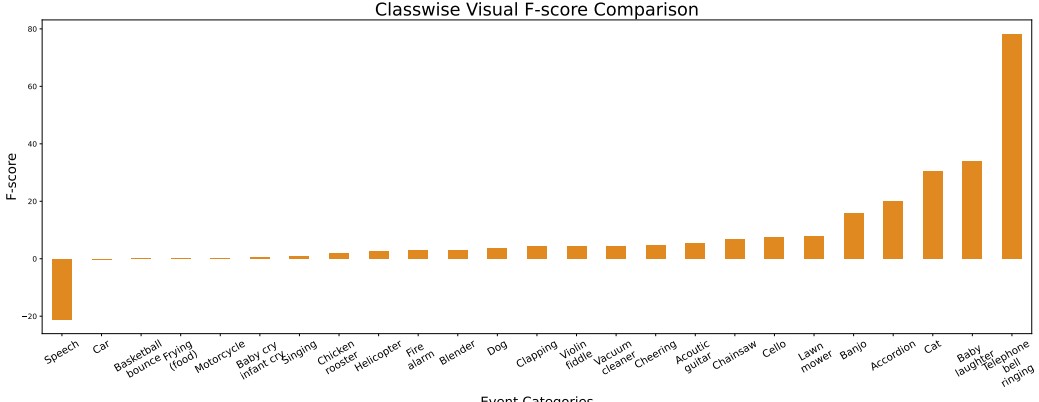

Figure 7: **Class-wise improvement on visual events.** VALOR's visual segment-level labels clearly outperforms the video-level labels. Note that CLIP is applicable to extract global but not fine-grained information from visual inputs. Thus, it is not expected to produce proper visual cues for the "Speech" event, which requires close attention to mouth movements.

## 6 Conclusion

We propose **V**isual-**A**udio **L**abel Elab**or**ation (VALOR) for weakly-supervised Audio-Visual Video Parsing. By harnessing large-scale pre-trained contrastive models CLIP and CLAP, we generate fine-grained temporal labels in audio and visual modalities, providing explicit supervision to guide the learning of AVVP models. We show that utilizing modality-independent pre-trained models (CLIP and CLAP) and generating modality-aware labels are essential for AVVP. VALOR outperforms all the previous works when comparing in a fair setting, demonstrating its effectiveness. In addition, we demonstrate the generalizability of our method in the Audio-Visual Event Localization task, where we improve the baseline greatly and achieve a state-of-the-art result.

**Limitations**   While VALOR performs well on AVVP, it is uncertain whether it will maintain this efficacy when the number of events to classify expands. Moreover, because CLIP is far from perfect at capturing fine-grained visual details, it may fail to generate precise labels when the subject of the event is small or when the video quality is poor, potentially confounding the model.

**Broader Impacts**   As an event recognition model, VALOR could be applied to future intelligent surveillance systems. While may reduce physical crime concerns, it could on the other hand infringe people's privacy and rights. Since the input consists of videos of people, data privacy issues are inevitable, and it is essential to prioritize data protection against unauthorized access.

## Acknowledgments and Disclosure of Funding

We thank National Center for High-performance Computing (NCHC) for providing computational and storage resources. We appreciate the NTU VLL members: Chi-Pin Huang, Kai-Po Chang, Chia-Hsiang Kao, and Yu-Hsuan Chen, for helpful discussions.

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

Table 6: **The List of Input Captions and Thresholds for CLIP and CLAP.** We add the prompt "A photo of" before each event name to make CLIP's input captions and the prompt "This is a sound of" to make CLAP's input captions.

| Events | Input Captions | | thresholds $\theta$ | |
| | CLIP | CLAP | $\theta^{\text{CLIP}}$ | $\theta^{\text{CLAP}}$ |
|---|---|---|---|---|
| Speech | A photo of people talking. | This is a sound of speech | 20 | 0 |
| Car | A photo of a car. | This is a sound of car | 15 | 0 |
| Cheering | A photo of people cheering. | This is a sound of cheering | 18 | 1 |
| Dog | A photo of a dog. | This is a sound of dog | 14 | 4 |
| Cat | A photo of a cat. | This is a sound of cat | 15 | 6 |
| Frying_(food) | A photo of frying food. | This is a sound of frying (food) | 18 | -2 |
| Basketball_bounce | A photo of people playing basketball. | This is a sound of basketball bounce | 18 | 4 |
| Fire_alarm | A photo of a fire alarm. | This is a sound of fire alarm | 15 | 4 |
| Chainsaw | A photo of a chainsaw. | This is a sound of chainsaw | 15 | 2 |
| Cello | A photo of a cello. | This is a sound of cello | 15 | 2 |
| Banjo | A photo of a banjo. | This is a sound of banjo | 15 | 2 |
| Singing | A photo of people singing. | This is a sound of singing | 18 | 1 |
| Chicken_rooster | A photo of a chicken or a rooster. | This is a sound of chicken, rooster | 15 | 2 |
| Violin_fiddle | A photo of a violin. | This is a sound of violin fiddle | 15 | 3 |
| Vacuum_cleaner | A photo of a vaccum cleaner. | This is a sound of vacuum cleaner | 15 | 0 |
| Baby_laughter | A photo of a laughing baby. | This is a sound of baby laughter | 15 | 2 |
| Accordion | A photo of an accordion. | This is a sound of accordion | 15 | 2 |
| Lawn_mower | A photo of a lawnmower. | This is a sound of lawn mower | 15 | 2 |
| Motorcycle | A photo of a motorcycle. | This is a sound of motorcycle | 15 | 0 |
| Helicopter | A photo of a helicopter. | This is a sound of helicopter | 16 | 2 |
| Acoustic_guitar | A photo of a acoustic guiter. | This is a sound of acoustic guitar | 14 | -1 |
| Telephone_bell_ringing | A photo of a ringing telephone. | This is a sound of telephone bell ringing | 15 | 2 |
| Baby_cry_infant_cry | A photo of a crying baby. | This is a sound of baby cry, infant cry | 15 | 3 |
| Blender | A photo of a blender. | This is a sound of blender | 15 | 3 |
| Clapping | A photo of hands clapping. | This is a sound of clapping | 18 | 0 |

# A    Caption Construction and Threshold Determination in VALOR

We provide detailed explanations on how we devise input captions for each event to be used with CLIP and CLAP. For the CLIP's input captions, we add the prompt "A photo of" before each event name and modify some of the captions to make them sound reasonable, *e.g.* changing "A photo of speech" to "A photo of people talking." As for CLAP, we add the prompt "This is a sound of" before each event name. All input captions devised for CLAP and CLIP are included in Table 6 for reference.

Furthermore, the determination of class-dependent threshold values, $\theta^{\text{CLIP}}$ for CLIP and $\theta^{\text{CLAP}}$ for CLAP, is based on the visual and audio segment-level F-scores of the validation split, respectively. These scores are achieved by comparing the segment-level pseudo labels generated by the respective models against the ground truth labels.

# B    More AVVP Implementation Details

In our experiments, we apply two different model architectures: 1) the standard model architecture, which is employed in VALOR, consists of a single HAN layer with a hidden dimension of 512; 2) the variant model architecture, which is used in VALOR+ and VALOR++, is a thinner yet deeper HAN model, comprising four HAN layers with a hidden dimension of 256. Both models contain approximately the same number of trainable parameters. The above details are summarized in Table 7. The models are trained using the AdamW optimizer, configured with $\beta_1 = 0.5$, $\beta_2 = 0.999$, and weight decay set to 0.001. We employ a learning rate scheduling approach that initiates with a linear warm-up phase over 10 epochs, rises to the peak learning rate, and then decays according to a cosine annealing schedule to the minimum learning rate. We set the batch size to 64 and train for 60 epochs in total. We clip the gradient norm at 1.0 during training. We attach the code containing our model and loss functions to the supplementary files.

# C    Additional Analysis

**More Details of Using Video Labels as Filters**    In this section, we provide more details regarding video label filtering. First, when video labels are not used as filters, we need to adjust the event thresholds again on the validation split. To save time for the adjustment, we transition from class-dependent thresholds to unified (class-independent) thresholds. This means that after the adjustment, the threshold for each event is the same. For the sake of fairness, we also switched to using unified thresholds when using video labels as filters. The experimental results, as shown in Table 8, indicate that simply changing the thresholds from class-dependent

Table 7: **Two Different HAN Model Architectures.** The "standard" model architecture is used in VALOR. The "variant" model architecture is used in VALOR+ and VALOR++.

| HAN model | standard | variant |
|---|---|---|
| *Model Arch. Hyper-parameters* | | |
| hidden dim | 512 | 256 |
| hidden layers | 1 | 4 |
| trainable params | 5.1M | 5.05M |
| *Training Hyper-parameters* | | |
| peak learning rate | 1e-4 | 3e-4 |
| min learning rate | 1e-6 | 3e-6 |

Table 8: **Ablation study of whether using video-level labels as filters.** Left: the ablation study of using video-level labels in **audio** label elaboration. Right: the ablation study of using video-level labels in **visual** label elaboration. To save the time required for tuning event thresholds, we have transformed class-dependent event thresholds into unified event thresholds, which means that the thresholds for each event are the same.

| Video labels as filters | Event Thresholds | Segment-level A | V | AV | Type | Event | Video labels as filters | Event Thresholds | Segment-level A | V | AV | Type | Event |
|---|---|---|---|---|---|---|---|---|---|---|---|---|---|
| ✗ | unified | 47.9 | 64.5 | 49.2 | 53.9 | 53.8 | ✗ | unified | **62.8** | 53.8 | 50.9 | 55.9 | 58.6 |
| ✔ | unified | **63.4** | 65.8 | 60.2 | 63.1 | **62.2** | ✔ | unified | 62.3 | 65.9 | 60.6 | 62.9 | 60.9 |
| ✔ | class-dependent | 62.7 | **66.3** | **61.0** | **63.4** | 61.8 | ✔ | class-dependent | 62.7 | **66.3** | **61.0** | **63.4** | **61.8** |

to unified does not significantly degrade the model's performance, whether in the audio or visual modality. However, if video label filtering is not applied, the resulting audio and visual pseudo labels become highly inaccurate, leading to a model with an audio F-score of only 47.9 and a visual F-score of only 53.8.

# D   VALOR with Pseudo Label Denoising

In this section, we explore the application of Pseudo Label Denoising (PLD), as proposed in VPLAN [96], to refine the segment-level labels generated by our method. The hyper-parameters for the PLD, specifically $K = 4$ and $\alpha = 6$ for the visual modality, and $K = 10$ and $\alpha = 10$ for the audio modality, are chosen based on the visual and audio F-scores on the validation split. From Table 9, we can see that PLD is less effective in refining our pseudo labels compared to VPLAN's pseudo labels ($+1.5$ v.s. $+2.22$ in segment-level metrics and $+2.28$ v.s. $+3.41$ in event-level metrics). However, it's worth noting the visual segment-level labels derived from our method **before** PLD are nearly as accurate as those from VPLAN **after** PLD (72.34 v.s. 72.51). Although we do implement PLD in the audio modality, no noticeable improvement is recorded for any audio pseudo labels. Referring to Table 10, the model trained with our denoised segment-level labels improves marginally. Nevertheless, we outperform VPLAN on Type@AV and Event@AV F-scores in segment-level and event-level metrics.

# E   Qualitative Comparison with Previous AVVP Works

Aside from quantitative comparison with previous AVVP works, we perform a qualitative evaluation as well. In Figure 8, we qualitatively compare with the baseline method HAN [73] and the state-of-the-art method JoMoLD [11]. In the top video example, JoMoLD erroneously predicts the "Speech" audio event, while all other methods accurately identify the audio events. In the bottom example, HAN produces identical temporal annotations for the "Speech" event in both modalities, despite the event only occurring audibly. Additionally, our method provides annotations that more closely align with the ground truth than either HAN or JoMoLD does when the events occur intermittently, which is challenging for models to generate accurate predictions.

# F   More Audio-Visual Event Localization Details

**Baseline Method**   We adopt the baseline model HAN to aggregate uni-modal and cross-modal temporal information as we have done in the AVVP task. For brevity, we introduce our baseline method from the procedure after feature aggregation. The segment-level audio features and visual features, $\hat{\boldsymbol{f}}_t^a$ and $\hat{\boldsymbol{f}}_t^v$ ($\in \mathbb{R}^d$), output from HAN are processed through a 2-layer feed-forward network (FFN) to yield the uni-modal segment-level predictions (logits), $\boldsymbol{z}_t^a$ and $\boldsymbol{z}_t^v$ ($\in \mathbb{R}^{(C+1)}$), respectively:

$$\boldsymbol{z}_t^m = \text{FFN}(\hat{\boldsymbol{f}}_t^m), \; m \in \{a, v\}, \tag{10}$$

Table 9: **PLD refinement.** We evaluate the fidelity (F-score) of the segment-level pseudo labels before and after pseudo label denoising (PLD). PLD is less effective in refining our pseudo labels compared to VPLAN's pseudo labels. However, the visual segment-level labels generated from our method **before** PLD are nearly as accurate as those generated from VPLAN **after** PLD (72.34 v.s. 72.51). Results are reported on the validation split.

| Methods | PLD | Audio | | Visual | |
| | | Seg | Event | Seg | Event |
| --- | --- | --- | --- | --- | --- |
| VALOR | ✗ | 80.78 | 71.69 | 72.34 | 66.36 |
| VALOR | ✔ | 80.78 | 71.69 | 73.84 (+1.5) | 68.64 (+2.28) |
| VPLAN [96] | ✗ | - | - | 70.29 | 64.68 |
| VPLAN [96] | ✔ | - | - | 72.51 (+2.22) | 68.09 (+3.41) |

Table 10: **Results of Training with Denoised Labels.** We outperform VPLAN on Type@AV and Event@AV F-scores in segment-level and event-level metrics with and without PLD. Results are reported on the testing split.

| Methods | PLD | Segment-level | | Event-level | |
| | | Type | Event | Type | Event |
| --- | --- | --- | --- | --- | --- |
| VALOR | ✗ | 62.0 | 61.5 | 56.7 | 54.2 |
| VALOR | ✔ | 62.2 | 61.9 | 56.6 | 53.7 |
| VPLAN [96] | ✗ | 61.2 | 59.4 | 54.7 | 50.8 |
| VPLAN [96] | ✔ | 62.0 | 60.1 | 55.6 | 51.3 |

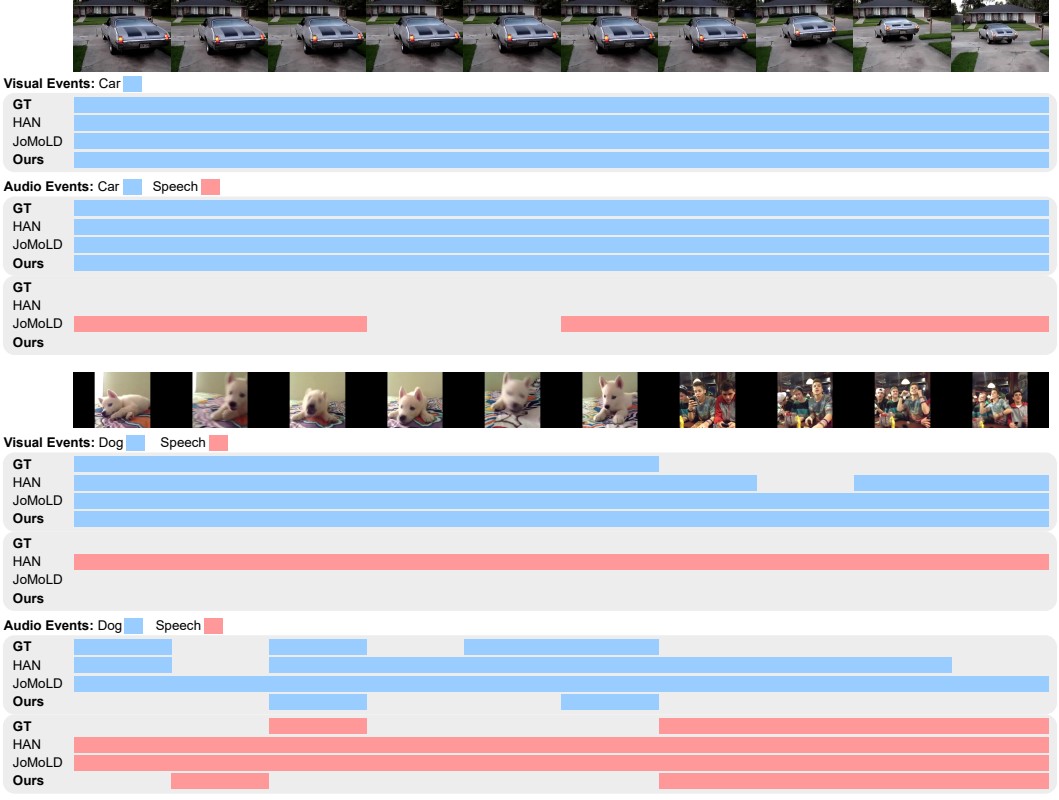

Figure 8: **Qualitative Comparison with Previous AVVP Works.** In general, the predictions generated by our method VALOR are more accurate than those produced by the other methods. "GT" denotes the ground truth annotations. We compare with HAN [73] and JoMoLD [11].

where $C + 1$ denotes the number of event classes and the "background" event. Since segment-level labels are not available in the weakly-supervised setting, we simply infer video-level logits $z \in \mathbb{R}^{C+1}$ by averaging all logits over time dimension $t$ and modality dimension $m$. Finally, the binary cross-entropy loss is applied to train

the model:

$$\mathcal{L}_{\text{video}}^{\text{ave}} = \text{BCE}(\text{Sigmoid}(\boldsymbol{z}), \boldsymbol{y}), \quad \boldsymbol{z} = \frac{1}{2T} \sum_t \sum_m \boldsymbol{z}_t^m \tag{11}$$

**Harvesting Training Signals**    The main idea of our method is to leverage large-scale open-vocabulary pre-trained models to provide modality-specific segment-level pseudo labels. We elaborate on how these pseudo labels are generated. Initially, segment-level audio logits and visual logits, $\boldsymbol{z}_t^{\text{CLAP}}$ and $\boldsymbol{z}_t^{\text{CLIP}}$ ($\in \mathbb{R}^C$), are generated in a manner identical to the AVVP task. Then, we use two sets of class-dependent thresholds, $\boldsymbol{\phi}^{\text{CLAP}}$ and $\boldsymbol{\phi}^{\text{CLIP}}$ ($\in \mathbb{R}^C$), to construct the uni-modal segment-level labels $\hat{\boldsymbol{y}}_t^a$ and $\hat{\boldsymbol{y}}_t^v$ ($\in \mathbb{R}^C$), respectively:

$$\hat{\boldsymbol{y}}_t^m = \boldsymbol{y} \wedge \{\boldsymbol{z}_t^P > \boldsymbol{\phi}^P\}, \ (m, P) \in \{(v, \text{CLIP}), (a, \text{CLAP})\} \tag{12}$$

In addition, we append an additional event "background" to the end of the segment-level labels $\hat{\boldsymbol{y}}_t^m$ to expand the dimension to $\mathbb{R}^{C+1}$. If $\hat{\boldsymbol{y}}_t^m$ consists solely of zeros, we assign the last dimension ("background") a value of one; otherwise, we assign it a value of zero. In other words, if an event could possibly occur in a video and the pre-trained model has a certain confidence that the event is present in a specific video segment, that segment will be labeled as containing the event; otherwise, the segment will be labeled as "background". Having prepared the segment-level pseudo labels $\hat{\boldsymbol{y}}_t^a$ and $\hat{\boldsymbol{y}}_t^v$, we compute binary cross-entropy loss in individual modality and combine them to optimize the whole model instead of using the video-level loss $\mathcal{L}_{\text{video}}^{\text{ave}}$:

$$\mathcal{L}_{\text{VALOR}}^{\text{ave}} = \text{BCE}(\text{Sigmoid}(\boldsymbol{z}_t^a), \hat{\boldsymbol{y}}_t^a) + \text{BCE}(\text{Sigmoid}(\boldsymbol{z}_t^v), \hat{\boldsymbol{y}}_t^v) \tag{13}$$

**Dataset & Evaluation Metrics**    The *Audio-Visual Event (AVE) Dataset* [72] is composed of 4143 10-second video clips from AudioSet [23] that cover 28 real-world event categories, such as human activities, musical instruments, vehicles, and animals. Each clip contains an event and is uniformly split into ten segments. Each segment is annotated with an event category if the event can be detected through both visual and auditory cues; otherwise, the segment is labeled as background. The AVE task is divided into a supervised setting and a weakly-supervised setting. In the former, we can obtain ground truth labels for each segment during training; in the latter, similar to the AVVP task setting, we can only obtain video-level labels. As with the AVVP task, we address the AVE task under the weakly-supervised setting. We follow [72] to split the AVE dataset into training, validation, and testing split and report the results on the testing split. Following the previous work [72], we use the accuracy of segment-level event category predictions as the evaluation metric.

**Implementation Details**    The pre-trained ViT-L CLIP and R(2+1)D are used to extract 2D and 3D visual features, respectively, which are then concatenated to represent low-level visual features. The pre-trained HTSAT-RoBERTa-fusion CLAP is used to extract audio features. We adopt the standard HAN model (1-layer and 512-dim) in this task and train the model with AdamW optimizer, configured with $\beta_1 = 0.5$, $\beta_1 = 0.999$, and weight decay set to $1e-3$. A learning rate scheduling of linear warm-up for 10 epochs to the peak learning rate of $3e-4$ and cosine annealing decay to the minimum learning rate of $3e-6$ is adopted. The batch size and the number of total training epochs are 16 and 120, respectively. We clip the gradient norm at 1.0 during training.

