# OpenReview forum: "Modality-Independent Teachers Meet Weakly-Supervised Audio-Visual Event Parser"
_NeurIPS.cc/2023/Conference — NeurIPS 2023 poster_

### Official Review · Reviewer_uN7g · 2023-07-03

**Soundness:** 3 good
**Presentation:** 3 good
**Contribution:** 3 good
**Rating:** 6
**Confidence:** 2

**Summary:**

This paper proposes a method called VALOR to assign segment level audio-visual (AV) event/object labels given weak labels at the video-level. It makes use of pseudo-labels obtained from unimodal pretrained models CLIP and CLAP to derive additional guidance for the AV model. The final model is trained using a combination of AV, audio-only and video-only losses, and the AV component is based on the hybrid attention network (HAN) structure.
Experiments are performed on two tasks, AV video parsing and AV event localization. In both cases, the VALOR approach outperformed previous studies on the respective tasks by a clear margin. Some additional ablation experiments further validate the design choices that have been made in the system.

* After rebuttal

Given the authors' response, and them clarifying my concerns, I am increasing my score, especially for presentation.

**Strengths:**

The main strength is that the experimental results clearly outperforms older approaches on the two tasks described in the paper.

* Originality: Even though the individual sub-components such as CLIp, CLAP, HAN are not novel, the paper brings them together in a certain way (unimodal guidance with dense labeling of the video segments) to solve the AV video parsing problem on a weakly labeled dataset.
* Quality: Experimental evaluations successfully support the use of the newly proposed VALOR technique.
* Clarity: Language is mostly clear.
* Significance: The paper probably establishes the new state-of-the-art the for the AV video parsing task on the LLP dataset.

**Weaknesses:**

* The main weakness of the paper is probably the flow of the paper.

1. There is a section called Preliminaries and then in Section 3.1, CLIP and CLAP are introduced. They can go to preliminaries.

2. The motivation behind adding Section 4.2 is not clear. Similar applications have been already included in the Introduction.
* Another weakness is lacking clarity about some of the experimental settings: Instead of Section 4.2, it could be better to save some space and describe the dataset a little clearer. For example,

1. how many labels are there, are they coming from a closed vocab? (We later get the answers to these questions by looking at Figs. 3 and 4)

2. Even though the timestamps are not given, do we know the order of the labels, or is it more like a bag of labels at the video level?
b. What does CLIP+CLAP without modality labels work in Table 2? Do we take the union of CLIP and CLAP labels and then use this as the ground truth for both audio and video losses?

* The improvements in F-scores are mostly stated between HAN and the VALOR approach, but as mentioned at times, there were other stronger baselines to compare (JoMoLD and CMBS in Tables 1 and 4, respectively). It could be better to mention improvements w.r.t. those ones although the gains will look slightly smaller in that case.


**Questions:**

1. Do we know the order of the weak labels?
2. In AV event localization evaluations, did the authors allowed a margin to evaluate the F-scores?
3. The paper describes Type@AV as the Average of A-only, V-only and AV F-scores. It might make it clearer if the name reflects this fact.
4. Event@ AV score is also not very self-explanatory. How does it differ from the AV F-score?
5. Do the authors consider how the number of occurrences of an event in the training dataset correlate with the final performance? Is there any bias do to class imbalance?
6. Paper mentions class-dependent threshold to binarize the CLIP/CLAP outputs. Are they manually or automatically tuned? If automatic, which subsets are used to determine these numbers?
7. Instead of binarizing the labels from CLIP/CLAP, have the authors tried to use the soft labels directly in the loss computation? Soft-labels have sometimes been useful in other semi/self-supervised applications.
8. What is the chosen segment length (duration)? Have the authors tested various granularity levels?
9. The notation is Section 2 can be improved. Even though we can understand their meaning from the context, there are many f's in various forms: $f$, $F$, $\mathrm{F}$
10. The level of detail in the case of MMIL looks a little long given that the rest of the paper does not mention it that frequently in the rest of the paper.

**Limitations:**

Some limitations have been mentioned in the paper.  For example, whether the model will be effective in the case of large vocabulary labels. Also how the inherent limitations of CLIP can harm the proposed VALOR approach.
1. It is not clear whether there are any issues due to class imbalance in the dataset. Or if the LLP data has a similar label distribution in train/dev/test splits?

---

> ### Author Rebuttal · Authors · 2023-08-10
>
> ## **Response to Reviewer uN7g**
> We thank Reviewer uN7g for the constructive comments and suggestive remarks. Please see our responses below for each raised issue.
>
> **Q1. Improvement of presentation flow. Sect. 2 and Sect. 3.1. The motivation behind adding Section 4.2.**
>
> Sect. 2 (Prelim) defines the task of AVVP, including the baseline HAN [72]. While Sect. 3 introduces our proposed model, Sect. 3.1 is to explain the idea of exploiting pre-trained cross-modality language models for zero-shot label transfer, in which CLIP and CLAP are utilized. As suggested by Reviewers 82wA and t3gt, we have verified that our method is not limited to the use of CLIP/CLAP.
>
> Our Section 1 explains audio-visual learning tasks and highlights the challenge of “modality unaligned” settings, which is also the main focus of our paper. To maintain the fluency and clarity of the Introduction, we only provide a few examples of related studies. As for Section 4 (Related Work), we provide thorough literature reviews covering relevant research on audio-visual learning in various settings.
>
> **Q2. Clarity in some experimental settings. How many labels? Are they from a closed vocab? Is the label order available? What’s CLIP+CLAP w/o modality labels in Table 2?**
>
> As noted in Sect. 5.1, there are 25 events (labels) in total, which are from a closed/predefined vocabulary. With the dataset and settings defined in [72], the order of the video labels is not available. Table 2 is for ablation study purposes. CLIP+CLAP without modality labels means the pseudo labels derived for each segment are the union of audio and visual labels produced by VALOR, followed by training the HAN model for AVVP. Since degraded performance was observed, it verifies the effectiveness of our VALOR in predicting modality-aware pseudo labels for cross-modal learning.
>
> **Q3. Possible improvements on strong baselines (JoMoLD and CMBS)?**
>
> In the second paragraph of Section 5.2, we have presented the improvement over JoMoLD. We note that VALOR scored higher across all metrics, including a 5.4 F-score improvement for segment-level Type@AV under a fair setting. In the quantitative results of Section 5.4, we also noted that we surpassed CMBS on the weakly-supervised AVE task with an improvement of 4.4% in accuracy.
>
>
> **Q4. Margin to evaluate the F-scores?**
>
> No, and we are not able to do so. This is because, according to [71], F-score is not the evaluation metric defined for the AVE task. Since the AVE dataset consists of video segments with equal and pre-determined length, we adopt accuracy as the evaluation metric Moreover, since each video contains only one event, each segment is labeled with only one event label, which is why accuracy is qualified as an evaluation metric.
>
> **Q5. Type@AV naming confusion**
>
> We thank the reviewer for the advice. We agree that Type@AV might not be sufficiently intuitive to represent the average of the A-only, V-only, and AV F-scores. However, since the term Type@AV was introduced in the first paper of the AVVP task [72]. Existing literatures (including ours) addressing AVVP follow this notation for presentation consistency.
>
> **Q6. How Event@AV differs from the AV F-score?**
>
> Defined by [72], Event@AV calculates the F-score of all audio/visual events (i.e., across all segments, modality-aligned or not); as for AV F-score, it is the F-score calculated for the “audio-visual” events only (i.e., across modality-aligned segments).
>
>
> **Q7. Does the number of event occurrences correlate with performance? Is there class imbalance bias?**
>
> To address this issue, we show Figure 2 in the rebuttal PDF, showing that the number of event occurrences does not correlate with the final performance on the validation split. Note that the leftmost sub-figure in Fig. 2 shows the number of videos containing each event, the middle sub-figure shows the event-wise audio F-scores, and the rightmost figure shows the event-wise visual F-scores. Moreover, since ㄍthe F-score is calculated by (2 * TP) / (2 * TP + FP + FN) for each class, its value would not be biased even with the presence of class imbalance.
>
> **Q8. Determination of class-dependent thresholds to binarize the CLIP/CLAP outputs**
>
> For simplicity and fairness, we select such thresholds for each class based on its best segment-level F-score on the validation split. As suggested by Reviewer veqt, if using a class-independent threshold, which is determined by the best overall segment-level F-score on the validation split, we observe a comparable performance in Table 1 of our global rebuttal pdf. Thus, the performance is not sensitive to the thresholds selected from the validation set.
>
> **Q9. Possible use of soft labels?**
>
> Yes, we have tried to let our model directly learn from the logits output from the pre-trained models (without thresholding into hard labels), which is the knowledge distillation (KD) experiment in the second row of Table 3. From this table, we observe that such labeling strategy only resulted in F-scores of 51.1/64.0 for audio/visual events, while our VALOR achieved 62.7/66.3. Thus, use of our proposed VALOR to leverage video-level labels for pseudo label prediction is desirable.
>
> **Q10. What is the chosen segment length? Have the authors tested various granularity levels?**
>
> Each segment length (duration) is 1 second. Since the length of each video segment is pre-determined and fixed by [72], we are not able to consider video segments at different granularity levels.
>
>
> **Q11. The notation in Section 2 can be improved (e.g., many f's in various forms: $f$, $F$, $\rm F$)**
>
> Thank you for pointing this out. $f\in R^d$ represents a segment audio/visual feature, $F\in R^{T \times d}$ represents the collection of all segment audio/visual features, and $\rm F\in R^{2 \times T \times d}$ represents the collection of all segment features in both modalities. To avoid confusion, we will replace $\rm F\in R^{2 \times T \times d}$ with $X$ instead.

---

> > ### Comment · Reviewer_uN7g · 2023-08-13
> >
> > I would like to thank the authors for providing detailed answers to my questions including the generation of graphs in the rebuttal pdf.
> >
> > In Q7, for class-imbalance vs. F-measure issue, I think the answer depends on how you exactly calculate the F-measure ((2 * TP) / (2 * TP + FP + FN) ) for multi-class classification. In one version, each of TP, FP, FN are the total sum over all classes, i.e. $ TP =  \sum_k TP_k $ where $ k$ is the class index (there are $K$ classes). Whereas in the other version, one computes the F-measure per class and then averages those across classes, i.e. $ F-measure  = \frac{1}{K} \sum_k F_k =  \frac{1}{K} \sum_k (2 * TP_k) / (2 * TP_k + FP_k + FN_k) $. Especially, in the second version, it is possible, for example, that a class with small number of samples will have a noisier F-measure estimate and that may skew the results. That was why I was interested in the class distribution. So, I disagree with the claim that "its value would not be biased even with the presence of class imbalance". However, the rebuttal shows that this is not the case, hence, there is not a problem.
> >
> > Overall, I am going to increase my score especially for presentation/clarity.

---

> > > ### Author Response · Authors · 2023-08-14
> > >
> > > We thank the reviewer again for the input and suggest further clarification. We agree that the class imbalance would still affect the F-measure, depending on how it is calculated. We will revise the description in the manuscript accordingly, so that it would not cause potential concerns.

---

### Official Review · Reviewer_t3gt · 2023-07-04

**Soundness:** 3 good
**Presentation:** 4 excellent
**Contribution:** 3 good
**Rating:** 7
**Confidence:** 4

**Summary:**

The paper aims to improve performance of audio-visual video parsing (AVVP) on the LLP dataset via a novel pseudo-labelling technique. This is because LLP contains only coarse video-level labels, and expects the model to match fine-grained, temporally dense labels during testing. This mismatch (due to the intense labelling effort that would be required to have temporally dense labels in the training set) is obviously an issue, and has been approached by prev. works using the Multi-modal Multiple Instance Learning (MMIL) loss  for soft-selection and label smoothing, without much success. This paper proposes to use CLIP and CLAP, two pre-trained image/audio encoders, to extract pseudo-labels for each video/audio in an attempt to create temporally dense labels for the training set. This work also uses CLIP and CLAP as feature encoders for better results. This method considerably outperforms previous works, and its design choices are motivated by an ablation study. The method also performs well on audio-visual event localization.

**Strengths:**

The paper is well written and does a good job at introducing the reader to the LLP task, the challenges it entails and why previous works have struggled with it. The figures are great, in particular figure 2, and the tables are all clear and have a clear purpose. The narrative of the paper is consistent and has good momentum, and the initial questions and issues it poses are adequately solved/answered by the end of the paper, with good justifications and conclusions.

The method is rather simple but quite elegant, and seems to be the a clear low-hanging fruit to improve the LLP task. The preliminaries are well-explained and the choice not to perform distillation (which, again, would seem like a low-hanging fruit) is well-justified.

Related work seems to be adequately, comprehensively, and fairly referenced and discussed.

When compared with other works, the results are very convincing. Ablations are also welcome and have clear motivations and conclusions. Table 3 is especially good in my opinion - it answers a lot of questions.





**Weaknesses:**

I don't think the method has any particularly noticeable weaknesses, apart from the limitations that are brought up after the conclusion, which are understandable and are largely outside of the scope of this work.

However, I think the paper would benefit heavily from more experiments. CLIP and CLAP are great, and they are compared with HAN (which is effectively a completely different model), but some comparisons with other audio/image encoders would help justify the choice of using CLIP and CLAP specifically. Also it would be great if the authors could experiment with different datasets or even new tasks, if this is possible, as it would give us a wider breadth of results to draw conclusions from.

Would be good to have the best numbers in bold for Table 2. Would be good to highlight second and third best to highlight valor and valor+ rather than just valor++.

Some extra comments on the poor performing classes in Figure 3 would be nice.

I would suggest that the authors focus more on the term "pseudo-labelling" rather than "teacher" since student-teacher modelling is a broad topic with multiple techniques, but clearly here the authors are performing pseudo-labelling, which is a much more specific technqiue.

**Questions:**

When do you plan to make the code available?

To the best of the authors' knowledge, are there any other works that use CLIP or CLAP for pseudo-labelling, perhaps in audio-only or image-only literature? If so, it would be very relevant to discuss such approaches in the related work.

**Limitations:**

Some limitations are mentioned, which is good. However, there should also be a broader impact discussion (can be brief) about the potential misuse of this technology (e.g., aiding automated audio-visual surveillance, what happens if this model is deployed and gives a wrong output, etc.).

---

> ### Author Rebuttal · Authors · 2023-08-10
>
> ## **Response to Reviewer t3gt**
> We thank Reviewer t3gt for the positive comments and suggestive remarks. Please see our responses below for each raised issue.
>
>
> **Q1. Possible to consider visual/audio-language models other than CLIP or CLAP? What about other datasets?**
>
> A1: In Table 2 of the rebuttal PDF, CLAP and CLIP are replaced by AudioCLIP and OpenCLIP, respectively. In the left subtable of Table 2, the audio performance of the model slightly drops when it is trained with the AudioCLIP’s generated labels. On the other hand, in the right subtable of Table 2, the visual performance of the model trained with OpenCLIP’s generated labels is nearly the same as that of the model trained with CLIP’s.
>
> Although most existing papers addressing the AVVP task only consider one dataset, we also tackled the task of audio-visual event localization using the AVE dataset [71], which is a dataset different from the LLP dataset for AVVP. The AVE experiment results are in Table 4 in the main paper, where we observed a 5.1 improvement in accuracy compared to the baseline method of using the HAN model. It is worth noting that, both AVE and AVVP tasks perform learning from audio-visual data in the challenging “modality unaligned” setting.
>
> **Q2. Better visualization for top cases/numbers in Table 2**
>
> A2: In Table 1 in the main paper, as suggested, the best numbers are highlighted in bold and the second best numbers are already underlined. We will take the suggestion and also highlight the third highest numbers and update the table caption accordingly.
>
> **Q3. Comments on poor performing classes in Figure 3**
>
> A3: When comparing to the model simply using videl-level labels for audio segment pseudo-labels, we found that “cat”, “baby cry” and “cheering” were the events with slightly degraded audio events, as shown in Fig. 3. For these three audio events, a drop up to 3 in F-score were observed. When performing error analysis, we found that this was mainly due to the decrease of true positives instead of the increase of false positive ones. This suggests that these three audio events exhibit large intra-class variations. Nevertheless, as also shown in Fig. 3, most audio events reported improved F-scores (up to 13).
>
> **Q4. Suggest focusing more on the term "pseudo-labeling" rather than "teacher"**
>
> A4: We agree that using the terms “teacher” and “student” might lead readers to think that we are employing the knowledge distillation techniques for teacher-student model learning. Since the focus of our work is to provide modality-specific segment-level labels by leveraging large-scale pre-trained open-vocabulary models (CLIP and CLAP), we will just use “pseudo labeling” in our paper to avoid confusion.
>
> **Q5. Will the code be publicly available?**
>
> A5: Definitely. We have enclosed the code regarding model architecture and loss calculation in the supplementary materials, and we will release the code within a week after the author notification.
>
> **Q6. Any other works that use CLIP or CLAP for pseudo-labeling, even for audio-only or image-only literature?**
>
> A6: As discussed in Sect. 4.1, VPLAN [95] addresses the AVVP task by utilizing CLIP for pseudo labeling, which leverages CLIP to generate visual segment-level labels. Recently,  LSLD [A] is proposed (after the NeurIPS submission deadline) to address the AVVP task with a similar method. Note that they only focus on visual pseudo labeling only, while we generate pseudo labels in both modalities. We will update Sect. 4 accordingly.
>
> [A] Fan et al. "Revisit weakly-supervised audio-visual video parsing from the language perspective." arXiv:2306.00595.
>
> **Q7. Discussion on potential misuse of this technology**
>
> A7: Since our VALOR requires pre-trained audio/visual-language models to predict pseudo-labels, the lack of ability in describing fine-grained or unseen semantic concepts would be the major limitation. Misclassification of audio or visual events due to the above constraint might endanger video analysis tasks like surveillance or autonomous driving. On the other hand, since our VALOR is not a generative model, we do not expect that its misuse would result in DeepFake-like deceptive content.

---

> > ### Comment · Reviewer_t3gt · 2023-08-12
> > **Response to authors**
> >
> > Thank you for answering my questions. You have clarified all my doubts. Very happy to hear that the code will be made available.
> >
> > Happy to raise my score slightly based on the convincing responses to all reviewers. I am not completely familiarized with this specific field in depth, which is why my confidence score is not 5. But overall, I think the paper is quite good.

---

> > > ### Author Response · Authors · 2023-08-13
> > >
> > > We are glad that our responses have sufficiently clarified your raised issues, which definitely help us strengthen our work. We will make the code available as promised.

---

### Official Review · Reviewer_Qd1s · 2023-07-07

**Soundness:** 3 good
**Presentation:** 3 good
**Contribution:** 3 good
**Rating:** 6
**Confidence:** 4

**Summary:**

The paper proposes visual-audio label elaboration (VALOR) for weakly supervised audio-visual video parsing. It generates fine-grained temporal labels in audio and visual modalities by harnessing large-scale pretrained contrastive models CLIP and CLAP and providing explicit supervision to guide the learning of AVVP models. The paper shows that utilizing modality-independent pretrained models and generating modality-aware labels are essential for AVVP.

**Strengths:**

1 The paper proposes a simple and effective AVVP framework, VALOR, to harvest modality and temporal labels directly from video-label annotations, with an absolute improvement of +8.0 F-score.

2 The paper is the first to point out that modality independence could be crucial for audio-visual learning in the unaligned and weakly-supervised setup.

3 VALOR achieves new state-of-the-art results with significant improvements on AVVP (+5.4 F-score) with generalization to AVE (+4.4 accuracy) jointly verified.

**Weaknesses:**

1 The paper is really well-written and clear. It would be great if the paper can have a wider use scenerio. The current paper and contribution, experiments are limited to LLP dataset. It would be great if the paper could extend the method to other datasets. Adopting other widely used datasets to validate the proposed method can be a big plus for the paper.

2 The paper achieves big improvements than previous methods. It would be great if the paper could add FLOPs, throughput and number of parameters comparisons with previous methods. Model complexity comparison is a common practice for video related methods.

**Questions:**

Please check [*Weaknesses]

---

> ### Author Rebuttal · Authors · 2023-08-10
>
> ## **Response to Reviewer Qd1s**
> We thank Reviewer Qd1s for the positive comments and suggestive remarks. Please see our responses below for each raised issue.
>
> **Q1. Extension to datasets other than LLP?**
>
> A1: In addition to audio-visual video parsing (AVVP), we also tackled the task of audio-visual event localization using the AVE dataset [71], which is a dataset different from the LLP dataset for AVVP. The AVE experiment results are in Table 4 in the main paper, where we observed a 5.1 improvement in accuracy compared to the baseline method of using the HAN model. It is worth noting that both AVE and AVVP tasks perform learning from audio-visual data in the challenging “modality unaligned” setting.
>
> **Q2. Details on FLOPs, throughput, and number of parameters when compared with previous methods.**
>
> A2: Please see table below for the implementation details and comparisons. We also add such information in Table 3 of the rebuttal PDF.
>
> |                        | VALOR    |VALOR+|  HAN |MM Pyramid| MGN | JoMoLD | CVCMS|
> |------------------------|:--------:|:----:|:----:|:--------:|----:|--------:|-----:|
> | FLOPs (K)↓             |   17.3   | 27.6 | 17.3 | 84273    |901.5|  17.3   | 98.9 |
> | throughput (K)↑        |   45.1   | 40.8 | 48.3 | 9.0      |16.2 |  46.8   | 14.7 |
> | trainable params (M)↓  |   5.1    | 5.0  | 4.6  | 44.0     | 4.4 |  4.6    | 11.4 |

---

> > ### Comment · Reviewer_Qd1s · 2023-08-17
> > **Thank you for the rebuttal!**
> >
> > Thank you for these results! I do not have further concern of the paper. It seems the paper and responses are good to all reviewers. Thus, I increase the score a little bit. Thanks for the contribution!

---

> > > ### Author Response · Authors · 2023-08-21
> > >
> > > We are glad that our responses have sufficiently cleared all your concerns. Your suggestions will definitely help us strengthen our work in the next revision.

---

### Official Review · Reviewer_JWv6 · 2023-07-12

**Soundness:** 4 excellent
**Presentation:** 3 good
**Contribution:** 3 good
**Rating:** 8
**Confidence:** 3

**Summary:**

The paper tackles the task of audio-visual event parsing, where the goal is to independently recognize the localize the events occurring in the visual and audio modality. The paper argues that modality independent processing can be crucial for this task compared to joint modeling of the two modalities. Consequently, the paper makes use of pre-trained vision and audio models like CLIP and CLAP to harvest temporal labels independently for each modality. The proposed method achieves clear boost in scores across the board. The experiments are clear and sufficient.

Rebuttal: I have read the rebuttal. The rebuttal answers all my questions. So I have raised my rating.

**Strengths:**

I think the paper is clearly written (in most places), has a clear motivation and proposes simple intuitive methods to solve the discussed issues. The improvement in performance is significant and is useful for future works in this space.

**Weaknesses:**

The proposed automatic label harvesting using CLIP and CLAP have not been evaluated directly. Since a test set is available with segment labels, is it possible to evaluate the automatic annotation technique using the ground-truth labels on these test sets to directly evaluate how well the automatic annotation procedure works?

The automatic training signal extraction technique could have been explored in more detail, as it is the crux of the paper. To start with line 156-157 are quite unclear. What does “contrastive models understanding logits” mean? The intersection operation with y is not described in the text at all. Also, how are the threshold values decided for each class? Is it helpful to keep a slightly lower confidence label (z_t < theta) that is present in “y”? If both CLIP and CLAP models predict the same label with low probability (because something else is dominating the scene), can we keep that? Also some captions that are used to query CLIP or CLAP could be too fine-grained. E.g. the exact musical instrument or vehicle (car or motorcycle) might be hard to identify, so during the automatic training label harvesting, multiple variations of the captions could have been tried, such as using synonyms for example.

**Questions:**

See weaknesses.

**Limitations:**

Yes, the authors have discussed the limitations.

---

> ### Author Rebuttal · Authors · 2023-08-10
>
> ## **Response to Reviewer JWv6**
> We thank Reviewer JWv6 for the positive comments and suggestive remarks. Please see our responses below for each raised issue.
>
> **Q1. How accurate are our pseudo labels (derived via CLIP/CLAP)?**
>
> A1: In Table 7 of the supplementary material, we have evaluated the quality of the pseudo labels generated by VALOR on the test split. As seen in Table 7, the audio segment-level F-score reached 84.92, and the visual segment-level F-score reached 82.8. These scores are significantly better than the approach directly leveraging video-level labels as segment labels, whose audio and visual segment-level F-scores are 79.33 and 69.30, respectively.
>
> In addition, we also evaluate the pseudo labels simply predicted by CLIP/CLAP (i.e., without our video label filtering), resulting in 17.74 and 31.89 F-scores in audio and visual modalities, respectively. This indicates the effectiveness of our VALOR which exploits video-level labels to guide the learning in each modality.
>
> **Q2. More details about the automatic training signal extraction technique. What does “contrastive models understanding logits” mean? The intersection operation with y is not described in the text at all.**
>
> A2: We thank the reviewer for giving us the opportunity to improve our paper. Regarding “contrastive models understanding logits”, they are the logits output from the pre-trained contrastive models (CLIP or CLAP), denoted as where superscript could represent CLIP or CLAP. This term comes from our use of pre-trained contrastive models for describing the associate segments, however, we will simply rephrase this term as modality-aware logits to avoid confusion.
>
> As for the intersection operation, it is the logical AND between the hard labels obtained from modality-aware logits (with thresholding) and the video-level labels. This operation eliminates the events that are erroneously predicted by the pre-trained models but not presented in the video. We will add this to the main paper for completeness.
>
>
> **Q3. How are the threshold values decided for each class? Possible to exploit labels with lower confidence (i.e., )? What if both CLIP and CLAP predict the same label but with low probability?**
>
> A3: For simplicity and fairness, we select such thresholds for each class based on its best segment-level F-score on the validation split. If using a class-independent threshold, which is determined by the best overall segment-level F-score on the validation split, we observe a comparable performance in Table 1 of our global rebuttal pdf. Thus, the performance is not sensitive to the thresholds selected from the validation set.
>
> Regarding the second issue, reducing the thresholds for events occurring in the video may not be helpful. This is because such an approach is equivalent to reducing all event thresholds in our method.  In our work, since such thresholds are selected via validation, any further adjustments to the thresholds would only result in poorer pseudo labels.
>
>
> **Q4. Possible to exploit captions (e.g., variants or synonyms) for CLIP or CLAP to improve pseudo label quality?**
>
> A4: Following the suggestion of the reviewer, we now utilize caption variants using different prefixes for CLIP/CLAP. For CLIP, the prefixes are 'A photo of', 'An image of', and 'This photo contains'; as for CLAP, the prefixes are 'This is the sound of' and 'This audio contains'.
>
> In addition, we also follow the suggestion and consider synonyms when describing event labels. For instance, synonyms for 'car' are 'automobile' and 'motorcar', synonyms for 'cat' are 'feline' and 'kitty', and synonyms for 'laughing baby' are 'chuckling infant' and 'giggling baby'.
>
> With the above caption manipulation/augmentation, we observe a 2.1 improvement in pseudo labels' visual F-score when employing three captions as opposed to a single caption per event. However, in terms of audio, this approach does not yield superior results and actually results in a 1.8 decrease in F-score. The complete results are presented in Table 4 of our rebuttal PDF. How to properly exploit captions for improved pseudo label prediction will be among our future research directions.

---

> > ### Comment · Reviewer_JWv6 · 2023-08-19
> >
> > Thank you for your detailed answers. I have raised my rating.

---

> > > ### Author Response · Authors · 2023-08-21
> > >
> > > We thank the reviewer again for the positive remarks. The discussions above will definitely help us strengthen our work.

---

### Official Review · Reviewer_82wA · 2023-07-14

**Soundness:** 2 fair
**Presentation:** 3 good
**Contribution:** 2 fair
**Rating:** 6
**Confidence:** 4

**Summary:**

This paper proposes methods for weakly-supervised audio-visual event learning. The method relies primarily on pre-trained audio-language (CLIP) and visual-language (CLAP) models to guide the learning process. These pre-trained models serve as teachers and provide pseudo-labels which are then used to compute loss which guides the overall learning process. Experiments are done on an LLP dataset and the proposed method improves the state of the art by a good margin.

**Strengths:**

– The proposed approach is simple – it relies on pre-trained CLIP and CLAP models and achieves good performance in both segment level and event level.

– One key claim the authors make is that modality independent learning can be crucial for audio-visual learning in some conditions. While most multimodal learning works focus on using multiple modalities to improve performance on a given task, the significance of learning independently from each modality is a good point to highlight.

– The paper is mostly clear and easy to follow except for some mathematical descriptions which I believe can be improved at a few places.


**Weaknesses:**

— The proposed approach relies heavily on the pre-trained models audio-language and image-language. Considering that it might be important to understand the impact of these models themselves. I believe that the paper is missing some crucial analysis in those areas. Some which might help pain a better picture.

--- What kind of information (pseudo-labels) these pre-trained models are providing for the audio and video modalities ? Can we get some insight into the distribution of \hat{y}^m_t and how that is related (if any) to the actual video level ? Do the audio and visual model provide complimentary or similar information ? This can be analyzed before any training.

--- It’s not fully clear (Table 3) what happens if the ground truth event labels are not used at all and just the pre-trained models are used to train.

--- What is the impact of using different pre-trained models -- different CLIP/CLAP type models?

— I think experiments on some other weakly supervised data might be helpful.

— While the “non alignment” of the modalities in the videos does sound like a valid problem to address – it is not clear how much of that is present in the LLP dataset. How often does the video and the audio do align with the event label ? There is no analysis of to what extent that is present in the current dataset and to what extent it gets addressed.

Some other comments/questions

-- What is KD in in Table 3 ? Is it the one where ground truth labels are not used ? Just knowledge distillation from teachers ?
-- Why is label smoothing applied to modality training targets ( line 106 ) but it’s not applied anywhere else ?
-- For event localization, I am not sure accuracy is the right metric. Some metric as well as visualization which factors in the beginning and of the event will be more informative.
-- The notations (line 145 - 147) q^P and q^m are confusing. Simplifying and clearly explaining what precisely they represent will improve clarity.
-- How are the threshold parameters \theta^P obtained for each class ?
-- In the current VALOR formulation only CLIP/CLAP outputs for only classes marked to be present are used (the logical and with y). What if you


--- Updated review after rebuttal----

**Questions:**

Please follow-up on the questions/concerns in the weakness section.

**Limitations:**

The authors describe limitations of their work in terms of whether the approach will generalize to larger settings. Societal impact and other such limitations are not discussed. Given the nature and scope of this paper that might be okay.

---

> ### Author Rebuttal · Authors · 2023-08-10
>
> ## **Response to Reviewer 82wA**
> We thank Reviewer 82wA for the positive comments and suggestive remarks. Please see our responses below for each raised issue.
>
>
> **Q1. Missing analysis on pre-trained audio/image-language models.**
>
> **(1a) What info is extracted from the pre-trained single-modality models?**
>
> A1a: The information produced by the pre-trained models is their confidence (i.e., logit) of the occurrence of each event in a video segment. The hard pseudo labels derived from these logits and video-label filtering indicate the presence of the events in each video segment.
>
> **(1b) Pseudo labels and their relation to the actual video labels.**
>
> A1b: In the supplementary material Table 7, we first validate the accuracy of the pseudo labels generated by VALOR before using them to train the model. The audio pseudo labels achieve an audio F-score of 84.92, and the visual pseudo labels achieve a visual F-score of 82.8. In contrast, if we use video labels directly as segment labels, the audio F-score is only 79.33, and the visual F-score is only 69.3. This demonstrates that the pseudo labels we generated are much more reliable.
>
> **(1c) Pseudo labels and their relation to the actual video labels.**
>
> A1c: In the supplementary material Table 7, we first validate the accuracy of the pseudo labels generated by VALOR before using them to train the model. The audio pseudo labels achieve an audio F-score of 84.92, and the visual pseudo labels achieve a visual F-score of 82.8. In contrast, if we use video labels directly as segment labels, the audio F-score is only 79.33, and the visual F-score is only 69.3. This demonstrates that the pseudo labels we generated are much more reliable.
>
> **(1d) What if no ground truth labels are used during training?**
>
> A1d: If the video-label filtering is not performed (i.e., discard the intersection operation with the video-level labels), significantly degraded performances would be expected because this hurts the pseudo label quality. We now provide such results as an additional ablation in the initial rows of the two subtables found in the rebuttal PDF Table 1. For example, the resulting audio/visual F-score is merely 47.9/53.8, compared to 63.4/65.9 produced by our VALOR.
>
> If the reviewer's intention is to suggest not only avoiding the use of video labels for filtering but also not using video labels as the ground truth for calculating the loss, then the task becomes purely unsupervised. This deviates too much from the original setting (weakly-supervised) and goes beyond the scope of our work.
>
> **(1e) What’s the impact of pre-trained models?**
>
> A1e: To further assess the flexibility of using pre-trained models in VALOR, we now replace CLAP and CLIP with AudioCLIP [24] and OpenCLIP [A], respectively. The experimental results of substituting CLAP with AudioCLIP are presented in the left subtable of Rebuttal PDF Table 2, while the results of replacing CLIP with OpenCLIP are shown in the right. The segment-level audio F-scores before and after the substitution are 63.4 and 61.0, respectively.The visual F-scores before and after the substitution are 62.3 and 61.6, respectively. We can conclude that our VALOR is not limited to the use of the specific instance of CLIP/CLAP.
>
> [A] Cherti et al., “Reproducible scaling laws for contrastive language-image learning.” In CVPR, 2023.
>
> **Q2. Missing analysis on how often visual-audio misalignment occurs**
>
> A2: To address this issue, we consider the validation split of LLP. We found that there are a total of 9126 event segments with labels assigned for at least one modality, and 4048 of them appear to be non-aligned, i.e., the label is assigned to exactly one modality. The baseline method, HAN [72], only correctly predicts 188 segments (accuracy of 4.6%). While our VALOR and VALOR++ were able to predict correct per-modality labels for 1518 and 1850 segments (accuracy of 37.5% and 45.7%), respectively. The SOTA method of JoMoLD only predicted 1471 of them and thus resulted in a poorer accuracy of 36.3%. Thus, it shows that our method is better at handling the “modality misalignment” problem.
>
> **Q3. Missing analysis on how often visual-audio misalignment occurs**
>
> A3: KD in Table 3 means knowledge distillation, indicating that the model learns directly from the logits output from CLIP and CLAP. The details of KD are mentioned in Sect. 3.2 in the main paper. Yes, the ground-truth video labels are not used in KD.
>
> **Q4. Why is label smoothing applied to modality training targets (line 106) but not anywhere else?**
>
> A4: Label smoothing is a heuristic method introduced by Tian et al. [72], which multiplies the ground-truth video labels by modality-dependent coefficients to assign labels to audio and visual data. However, [72] did not explicitly explain how such coefficients are determined. On the other hand, our method directly predicts pseudo labels in each modality by utilizing pre-trained audio/visual-language models, under the guidance of video-level labels.
>
>
> **Q5. Proper metric for event localization**
>
> A5: In the AVE task determined by [71], each video is divided into segments of equal and fixed length, with each segment containing only one event label. With such a setting, only per-segment prediction accuracy is required for evaluation. Note that accuracy is the official metrics [71] used in the AVE task.
>
> **Q6. Confusing notations (line 145 - 147) of $q^P$ and $q^m$**
>
> A6: $q$ represents the probabilities/logits of event labels. The superscript $P$ denotes the output from pre-trained models (CLIP or CLAP) and $m$ denotes the data modality (audio or visual). For example, $q^{CLIP}$ is the probabilities output from CLIP and $q^v$ denotes the visual probabilities output from the HAN model.
>
> **Q7. Determination of the threshold parameters $\theta^{P}$**
>
> A7: For simplicity and fairness, we select such thresholds for each class based on the performance on the validation split, i.e., the best segment-level F-score.

---

> > ### Comment · Reviewer_82wA · 2023-08-15
> >
> > Thanks for the detailed rebuttal. Several of the my concerns were addressed. I have made my overall score more positive.

---

> > > ### Author Response · Authors · 2023-08-15
> > >
> > > We are glad that our responses have sufficiently clarified your raised issues. Your comments and suggestions are greatly appreciated, which definitely help us strengthen our work.

---

### Official Review · Reviewer_veqt · 2023-07-17

**Soundness:** 3 good
**Presentation:** 3 good
**Contribution:** 3 good
**Rating:** 6
**Confidence:** 5

**Summary:**

In this work, the authors propose a unified weakly supervised audio-visual scene understanding framework for audio-visual video parsing and audio-visual event localization. Different from previous works, the proposed Visual-Audio Label Elaboration (VALOR) method is simple and effective. It leverages large-scale audio-text and visual-text contrastively pre-trained models as the modality teachers to predict individual labels for audio and visual modalities to tackle the modality and temporal uncertainty issue and boost event parsing performance.  Extensive experiments and ablation studies on the LLP and AVE datasets can validate the effectiveness of the proposed approach.

**Strengths:**

+ The proposed method is new and technically sound.
To alleviate drawbacks in past approaches, the proposed VALOR leverages large-scale audio-text and visual-text contrastively pre-trained models as the modality teachers to predict individual labels for audio and visual modalities for weakly-supervised audio-visual event parsing tasks.

+ Extensive experiments and ablation studies on two datasets are provided, and the strong results can demonstrate the effectiveness of the proposed method.

+ The core code is provided in the supplementary material, and the authors promised to release the source code.

+ The paper is easy to follow.

**Weaknesses:**

+ The process for selecting class-dependent thresholds in VALOR is not clearly outlined in the paper. These thresholds are crucial hyperparameters for generating audio and visual labels for training, and it appears that different classes may require different thresholds. Further explanation is needed on how these thresholds are chosen and how changes in these thresholds could potentially impact the performance of event parsing.

+ The authors did not provide a clear explanation as to why directly using Knowledge Distillation (KD) with audio and visual semantic class embeddings results in even worse parsing performance (especially for audio event parsing) on the LLP dataset, as shown in Table 3. While the proposed VALOR method does improve performance, a more detailed analysis on the KD models would be beneficial for further understanding.

+ The CLIP model uses a text prompt that is created by adding a "A photo of" prefix to the event's natural language form. However, the visual data involved in audio-visual event parsing are videos, not single images. This raises the question of whether there is a better prompt that could be used to improve model performance by taking into account the video nature of the data.



**Questions:**

Please address questions in Weaknesses.

**Limitations:**

The authors discussed method limitations in the paper.

---

> ### Author Rebuttal · Authors · 2023-08-10
>
> ## **Response to Reviewer veqt**
> We thank Reviewer veqt for the positive comments and suggestive remarks. Please see our responses below for each raised issue.
>
>
> **Q1. Selection of class-dependent thresholds in VALOR. What if other threshold choices?**
>
> A1: For simplicity and fairness, we select such thresholds for each class based on its best segment-level F-score on the validation split. If using a class-independent threshold, which is determined by the best overall segment-level F-score on the validation split, we observe a comparable performance in Table 1 of our global rebuttal pdf.  Thus, whether the thresholds are class-dependent or class-independent does not affect the results significantly.
>
>
> | CLAP Event Thresholds| Segment-level Audio F-score |
> |----------------------|:--------:|
> | class-dependent      |    62.7    |
> | class-independent  |   63.4   |
>
> | CLAP Event Thresholds| Segment-level Audio F-score |
> |-|:-:|
> | class-dependent |    66.3    |
> | class-independent |   65.9   |
>
>
> **Q2. Why Knowledge Distillation (KD) with audio and visual semantic class embeddings results in worse parsing performance?**
>
> A2: In order to perform knowledge distillation (KD), the Softmax function is applied to the logits predicted by pre-trained models (CLAP/CLIP) and those produced by the HAN model, followed by calculating their KL-divergence. Since the Softmax function predicts a single dominant label, which does not align with the multi-label setting of the AVVP task, the use of the KD-trained model is expected to obtain degraded performances than VALOR.
>
>
> **Q3. Possible to exploit visual prompts to describe visual information in videos?**
>
> A3: We note that visual/video data in the task of AVVP are presented as a collection of consecutive one-second video clips, with very few visual changes in each clip. Thus, it is sufficient to apply an image-text pre-trained model of CLIP with the standard prompt of “A photo of” to extract the visual information.
> Although we did not design prompts specifically for video data, we have created a variety of prompts tailored for images as suggested by Reviewer JWv6. Through this, we aimed to explore whether generating more prompts for each event could lead to more accurate pseudo labels. The experimental results are presented in Table 4 of the rebuttal PDF. We observe that generating more prompts for each event slightly improves the accuracy of the generated visual pseudo labels, resulting in a 2.1 increase in segment-level visual F-score. However, this approach does not improve the accuracy of the generated audio pseudo labels; instead, there is a slight decrease of 1.8 in segment-level audio F-score.

---

> > ### Comment · Reviewer_veqt · 2023-08-14
> > **Response to Authors**
> >
> > The rebuttal can address my questions. I will keep my positive rating. Thanks!

---

> > > ### Author Response · Authors · 2023-08-15
> > >
> > > We thank the reviewer for the positive remarks. We also appreciate the opportunity to clarify the raised issues, which definitely help us strengthen our work.

---

### Author Rebuttal · Authors · 2023-08-10

## **General Response**

We sincerely appreciate the valuable time and insightful feedback provided by the reviewers. We are grateful for the opportunity to address the concerns raised by each reviewer, which fundamentally strengthens our work. The strengths pointed out by the reviewers include:

**Method**: Motivation is clear. [Reviewer JWv6, t3gt]. Our VALOR is simple, technically sound, and novel. [Reviewer veqt, 82wA, JWv6, Qd1s, t3gt, uN7g].

**Experiment**: Extensive experiments (on two types of audio-visual tasks) and ablation studies are provided. [Reviewer veqt]

**Performance**: Strong performance was shown to prove the effectiveness of the proposed method. [Reviewer veqt, 82wA(good performance), JWv6, Qd1s, t3gt, uN7g]

**Presentation**: The paper is mostly clear and easy to follow. [Reviewer veqt, 82wA, JWv6, t3gt]

**Importance of modality-independent learning**: The claim that modality-independent learning being crucial for audio-visual learning under noisy audio-visual conditions is a good point to highlight. [Reviewer 82wA, Qd1s]

We would like to point out that particular concerns are raised, as listed below. Please refer to the responses to each reviewer for further details.

1. Selection of class-dependent thresholds [Reviewer veqt, JWv6, uN7g]
2. Variations of text prompt [Reviewer veqt, JWv6]
3. Ablation studies on video-label filtering [Reviewer 82wA]
4. Flexibility of re-trained model selection [Reviewer 82wA, t3gt]
5. Performance on runtime/speed [Reviewer Qd1s]
6. Label distribution analysis and its effect [Reviewer uN7g]


We thank the reviewers again for the suggestions and the raised issues. Should the reviewers have follow-up questions, we will be more than happy to answer them in the next phase. Given the recognized strengths in the initial reviews, together with additional experiments and clarification provided during rebuttal, we hope this work would be of great value to the audio-visual learning community.

---

### Decision · Program_Chairs · 2023-09-21

**Decision:**

Accept (poster)

**Comment:**

The paper has been well received with all reviewers being convinced by the rebuttal and being positive about the paper. The evaluation and additional comparisons provided (for instance, additional models other than CLIP and CLAP), the evaluation on the different settings were all appreciated. The consensus of the reviewers is to accept the work. AC agrees with this recommendation with the suggestion that the final version should include all the additional responses provided in the rebuttal.